# Prolonged tau clearance and stress vulnerability rescue by pharmacological activation of autophagy in tauopathy neurons

M. Catarina Silva [1], Ghata A. Nandi [1], Sharon Tentarelli[2], Ian K. Gurrell[3], Tanguy Jamier[3], Diane Lucente[4], Bradford C. Dickerson[5], Dean G. Brown[6], Nicholas J. Brandon[7] & Stephen J. Haggarty [1✉]

Tauopathies are neurodegenerative diseases associated with accumulation of abnormal tau protein in the brain. Patient iPSC-derived neuronal cell models replicate disease-relevant phenotypes ex vivo that can be pharmacologically targeted for drug discovery. Here, we explored autophagy as a mechanism to reduce tau burden in human neurons and, from a small-molecule screen, identify the mTOR inhibitors OSI-027, AZD2014 and AZD8055. These compounds are more potent than rapamycin, and robustly downregulate phosphory-lated and insoluble tau, consequently reducing tau-mediated neuronal stress vulnerability. MTORC1 inhibition and autophagy activity are directly linked to tau clearance. Notably, single-dose treatment followed by washout leads to a prolonged reduction of tau levels and toxicity for 12 days, which is mirrored by a sustained effect on mTORC1 inhibition and autophagy. This new insight into the pharmacodynamics of mTOR inhibitors in regulation of neuronal autophagy may contribute to development of therapies for tauopathies.

[1] Chemical Neurobiology Laboratory, Center for Genomic Medicine, Department of Neurology, Massachusetts General Hospital and Harvard Medical School, 185 Cambridge St CPZN 5400, Boston, MA 02114, USA. [2] Chemistry, Oncology R&D, AstraZeneca, 35 Gatehouse Dr, Waltham, MA 02451, USA. [3] Neuroscience, BioPharmaceuticals R&D, AstraZeneca, Cambridge, UK. [4] Molecular Neurogenetics Unit, Center for Genomic Medicine, Massachusetts General Hospital and Harvard Medical School, 185 Cambridge St CPZN, RM 5820, Boston, MA 02114, USA. [5] MGH Frontotemporal Disorders Unit, Gerontology Research Unit, Alzheimer's Disease Research Center, Department of Neurology, Massachusetts General Hospital and Harvard Medical School, 149 13th St. Suite 2691, Charlestown, MA 02129, USA. [6] Discovery Sciences, BioPharmaceuticals R&D, AstraZeneca, Waltham, MA, USA. [7] Neuroscience, BioPharmaceuticals R&D, AstraZeneca, Waltham, MA, USA. ✉email: shaggarty@mgh.harvard.edu

Tauopathies are neurogenerative diseases characterized by dysfunction and accumulation of abnormal tau protein in neurons and glia of affected brain regions[1], and include frontotemporal dementia (FTD), progressive supranuclear palsy (PSP) and Alzheimer's disease (AD), that can be either sporadic or inherited when caused by mutations in the *MAPT* gene encoding tau[2]. There are still no effective disease-modifying therapies and few experimental drugs focused on tau have reached clinical trials. One major factor contributing to this limited progress is the fact that the molecular mechanisms leading to neuronal death, and therefore potential therapeutic targets, are still not fully understood[3–5]. Accumulating evidence suggests that early tau mislocalization, oligomerization, and changes in solubility are better correlated with toxicity than later stage highly ordered tau filaments[4,6,7]. Therefore, early tau clearance may help our understanding of disease etiology and be a promising therapeutic strategy.

Another pathological hallmark of tauopathy is dysfunction of the autophagy-lysosomal pathway (ALP)[8,9]. Autophagy plays a key role in removal of aggregated proteins[10–12], and it appears to be a primary route of clearance for tau in healthy neurons[13]. Whether autophagy impairment is a contributor or a consequence of tauopathy is unclear[14,15]. Studies have shown evidence of abnormal ALP function in the brain of tauopathy patients, as well as in animal and cellular models, where accumulation of autophagic vesicles, lysosomes, and tau correlate with neuronal toxicity[9,16–22]. In these models, autophagy activators reduce the levels of misfolded and aggregated proteins, mitigating the spreading of tau and neuronal loss[10,22–28], supporting autophagy modulators' therapeutic potential[5,12,14,22,28–31].

Based on the hypothesis that autophagy is a disease-relevant therapeutic target, our working model focuses on pharmacological enhancement of ALP function in a disease context, to promote tau clearance. We performed a small-molecule screen to identify compounds that promote autophagy clearance of tau and rescue disease-relevant phenotypes in tauopathy patient-derived neurons. We identified three ATP-competitive mTOR kinase inhibitors (mTORi), OSI-027, AZD8055, and AZD2014 that show 100- to 1000-fold selectivity over class I PI3Ks (phosphatidylinositol 3-kinases)[10,32–36]. In tauopathy neuronal models, we demonstrate drug mechanism-of-action through mTOR complex 1 (mTORC1) inactivation, in direct correlation with autophagy activation and tau clearance (Supplementary Fig. 1). Most notably, we discovered that a single-dose 24 h treatment caused persistent reduction of tau for 12 days, resulting in a sustained effect on neuronal resistance to stress. Therefore, our results support a therapeutic potential for mTORC1 inhibitors in tauopathy-associated neurodegenerative disorders.

## Results

**Rationale for targeting ALP function in tauopathy neurons.** Evidence suggests that autophagy impairment is a hallmark of proteinopathies[16–21]. To investigate this, we employed human induced pluripotent stem cell (iPSC)-derived neural progenitor cells (NPCs) subsequently differentiated into neurons[21,37]. These cells were derived from unaffected tau wild-type (WT) individuals (Control-1, Control-2), from a PSP patient with a tau-A152T risk variant, and from a patient with FTD carrier of a tau-P301L autosomal dominant mutation[21,37,38]. In these patient-derived cell models that express tau at endogenous levels and recapitulate disease-relevant phenotypes[21,37], we measured ALP markers including the substrate selection and autophagosome biogenesis protein LC3-II, the lysosome-associated membrane glycoproteins LAMP1 and LAMP2, and the ubiquitin-binding autophagy receptor protein p62 (Supplementary Fig. 1a, Supplementary

Fig. 2a, e–h). In a time-course between 1 and 12 weeks of neuronal differentiation, we observed upregulation of these markers in tauopathy neurons (tau-A152T, tau-P301L), relative to controls. In parallel we observed upregulation of tau and accumulation of monomeric and high MW oligomeric phospho-tau (P-tau, Supplementary Fig. 2a–d). Antibody specificity for tau by western blot was verified by employing CRISPR/Cas9-engineered, polyclonal *MAPT* knocked-down cell lines (*MAPT*-Kd1-3), derived from the tau-A152T line[21], that show ~80% reduction in *MAPT* gene and protein expression (Supplementary Fig. 2i–k). Undifferentiated NPCs from the same parental line were employed as a control for no tau expression. Tau-specific bands were present in control neurons, showed increased intensity in FTD neurons and reduced intensity in *MAPT*-Kd neurons, and were absent in NPCs. These bands, corresponding to monomeric (~50–60 kDa) and >250 kDa oligomeric tau (Supplementary Fig. 2j) were employed from here on for western blot densitometry.

Together with our previous findings of decreased tau solubility and increased vulnerability to stress in FTD-derived neurons[21], these results showed tauopathy-associated disruption of autophagy. Based on the hypothesis that pharmacological enhancement of autophagy clearance of tau can rescue tauopathy phenotypes, and to further advance the concept that patient-derived cellular models can provide a valuable and clinically-relevant tool for drug discovery[39,40], we performed a small-molecule screen to identify activators of autophagy in human ex vivo neurons.

**Identification of activators of autophagy in human NPCs and neurons.** We conducted a small molecule microscopy-based screen in tau-WT NPCs, followed by validation of the hits in neurons, to identify autophagy activators independent of disease context (Fig. 1). Autophagy was measured based on number of fluorescent vesicles labeled with LysoTracker (red) and CYTO-ID (green) dyes. LysoTracker detects acidic organelles (lysosomes), whereas CYTO-ID is incorporated into autophagic vesicles (pre-autophagosomes, autophagosomes, phagolysosomes) with negligible staining of lysosomes. For assay validation, we used the selective mTORC1 inhibitor rapamycin[41,42] and the inhibitor of autophagosome-lysosome fusion, chloroquine (CQ)[43] (Supplementary Fig. 1a). Upon 24 h treatment, 5 μM rapamycin increased the number of LysoTracker+ and CYTO-ID+ vesicles (Supplementary Fig. 2l). A low concentration of 0.5 μM CQ, which alone did not have a measurable effect on LysoTracker or CYTO-ID, when combined with rapamycin caused further increase in the number of fluorescent vesicles compared with rapamycin alone (Supplementary Fig. 2l). This is the result of autophagy activation by rapamycin with simultaneous CQ blockage of autophagosome-lysosome fusion and clearance. Because in healthy cells autophagosomes are quickly eliminated by fusion with lysosomes without buildup of autophagic vesicles, combination with low dose CQ allowed real-time live-cell imaging and measurement of increased autophagy upon compound treatment.

We screened a chemogenomic library of 240 experimental and FDA-approved drugs known or predicted to activate autophagy. NPCs were treated for 24 h with compound at two doses (1 μM, 10 μM, +0.5 μM CQ) and positive hits (74) were identified as compounds that increased the number LysoTracker+ and CYTO-ID+ vesicles, relative to vehicle-treated NPCs, without obvious toxicity (visual inspection of cell loss). Dose-response curves for lead compounds showed different autophagy activation strengths, with rapamycin generating lower vesicle numbers (Fig. 1a) relative to OSI-027, AZD2014, and AZD8055 (Fig. 1b–d). The overlap between LysoTracker and CYTO-ID curves (Fig. 1a–d)

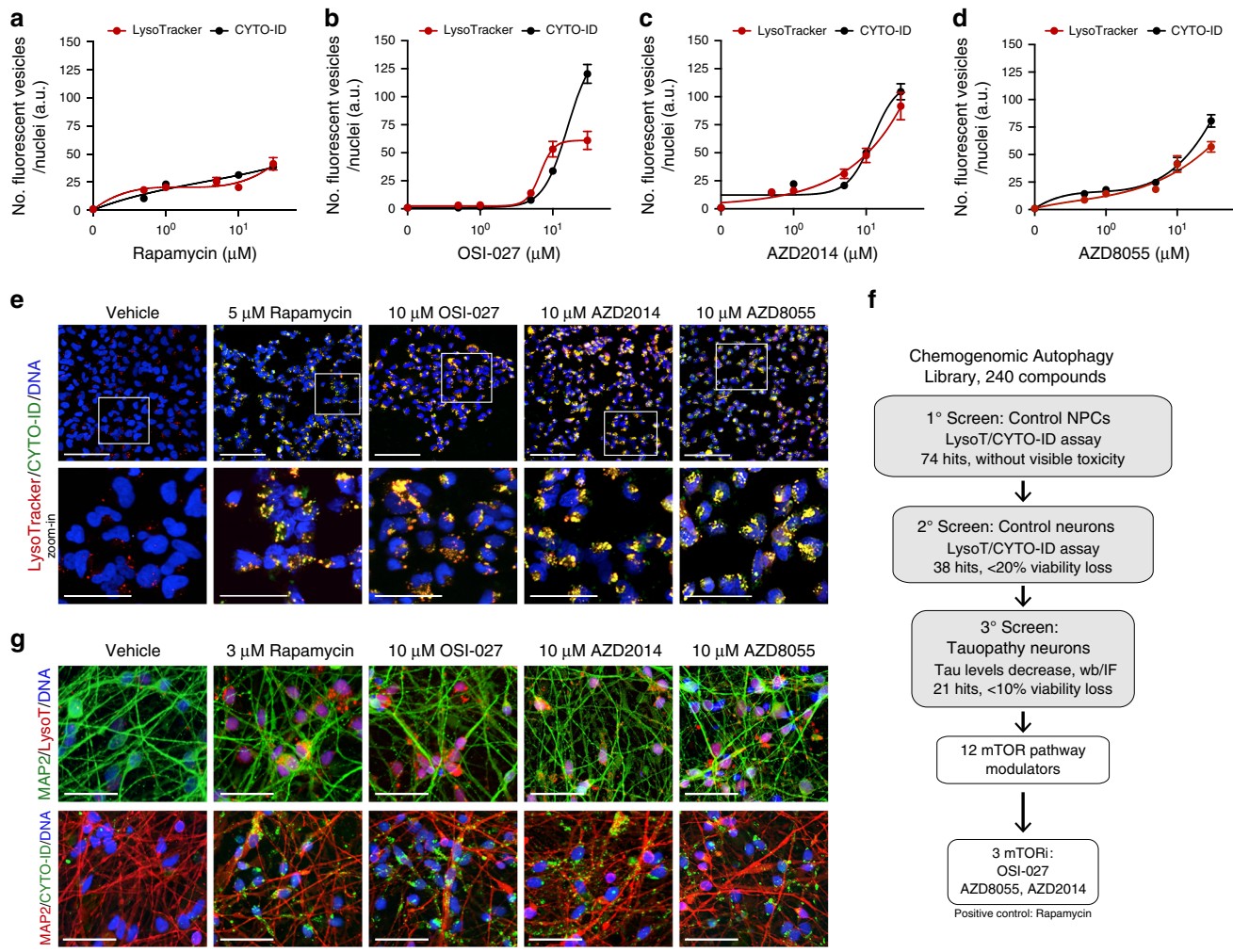

**Fig. 1 Microscopy screen for compounds that activate autophagy in human Control-1 NPCs and neurons. a–d** Compound dose-effect on number of LysoTracker[+] and CYTO-ID[+] fluorescent vesicles in NPCs (8330-8-RC1), after 24 h treatment with rapamycin, OSI-027, AZD2014, and AZD8055 (+500 nM CQ). Data points correspond to mean number of fluorescent vesicles per nuclei ± SEM from $n = 5$ biological replicates. **e** NPCs (8330-8-RC1) treated with compounds for 24 h at a dose that promoted maximum vesicle formation without toxicity. Scale bars are 50 μm. Representative images from $n = 5$ biological replicates. Inserts correspond to zoomed-in images below (scale bars are 25μm). **f** Screening strategy followed. **g** Representative images of 5-week differentiated neurons (8330-8-RC1) treated with compounds for 24 h (+500 nM CQ). LysoTracker[+] (top) and CYTO-ID[+] (bottom) vesicles are shown in the background of MAP2 neuronal staining ($n = 3$ biological replicates). Scale bars are 25 μm. Source data are provided as a Source Data file.

indicated a dose-dependent formation of autophagic vesicles (CYTO-ID[+]) and fusion with lysosomes (LysoTracker[+]). Representative cell images for lead compounds, at the dose where maximum number of vesicles was detected without toxicity, are shown in Fig. 1e. As a secondary screen, to identify activators of autophagy in neurons (Fig. 1f), compounds were tested in the same tau-WT cell line after 5 weeks of neuronal differentiation. We identified 38 compounds that activated autophagy in neurons with less than 20% reduction in viability (Fig. 1g).

**mTORi reduce tau burden in FTD patient-derived neurons.** To determine whether lead autophagy activators identified in control neurons were effective in tauopathy neurons and promoted tau clearance, FTD patient-derived neurons expressing tau-A152T[21,31] or tau-P301L[37,44] were employed (Fig. 1f). Neurons were treated for 24 h (no CQ) and examined for effects on total tau and P-tau[S396] by western blot. We identified 21 hits (<10% reduction in neuronal viability), of which 12 compounds are predicted to be mTOR pathway modulators with three compounds described as direct ATP-competitive mTORi. The dual catalytic mTOR complex 1 and complex 2 (mTORC1/2)

inhibitors AZD8055, AZD2014 and OSI-027 were further studied, with the mTORC1 allosteric inhibitor rapamycin used as control (Supplementary Table 1, Supplementary Fig. 1a).

To study dose-effect of mTORi on tau, A152T neurons differentiated for 6 weeks were treated with 0.1 μM–10 μM compound for 24 h and the effect on total tau and P-tau[S396] was measured by western blot (Fig. 2a–d). Compounds showed variability in dose-response, particularly at lower concentrations, but OSI-027 (Fig. 2b), AZD2014 (Fig. 2c) and AZD8055 (Fig. 2d) promoted stronger tau reduction than rapamycin (Fig. 2a). The concentration at which maximum reduction of tau was achieved, under assay conditions not affecting viability, was 10 μM for each compound (3 μM for rapamycin), with a ~80% downregulation of tau and P-tau[S396] for OSI-027 and AZD2014, and ~60% for AZD8055. The effective concentration at which 50% tau reduction was achieved (EC$_{50}$) was 100 nM for AZD8055 and AZD2014, and 1 μM for OSI-027. The same experiment was repeated across a larger dose range (1 nM–10 μM) and tau protein levels were measured by ELISA (Fig. 2e–h). Compounds' relative efficacy was corroborated, again with some variability at lower mTORi concentrations. Maximum reduction of tau and P-tau[S396]

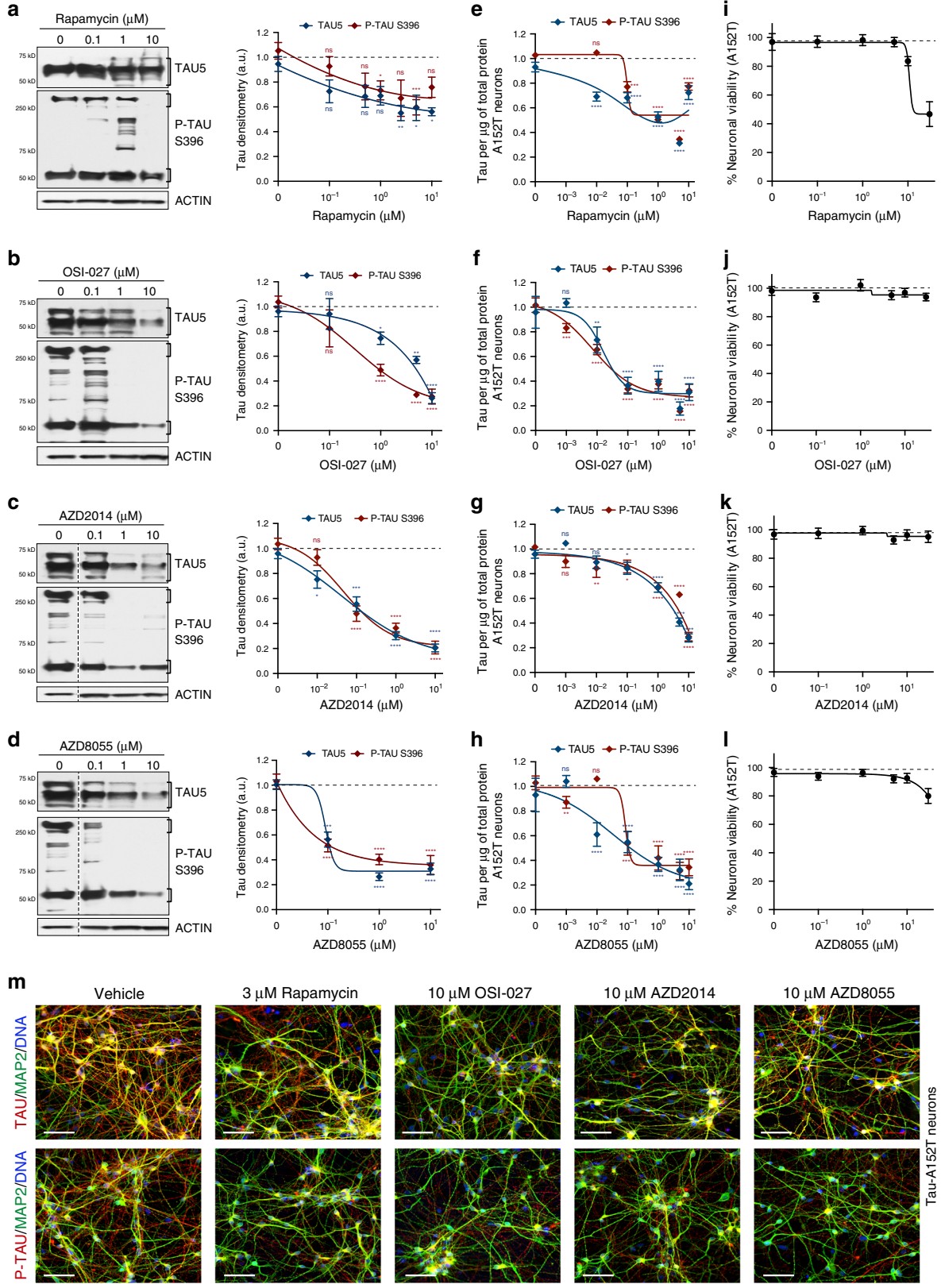

was observed at 10 μM of each mTORi (3 μM for rapamycin), with ≥70% tau reduction. These results suggest that mTORC1/2 inhibitors were more effective at promoting tau clearance than rapamycin, including clearance of high MW, oligomeric forms of P-tau$^{S396}$ (Fig. 2a–d). Moreover, for the concentrations tested, 24 h treatment of A152T neurons did not affect cell viability

(Fig. 2i–l), except for a ~20% decrease by AZD8055 at 30 μM. Only rapamycin became toxic at concentrations ≥5 μM, leading to 50% loss of viability (Fig. 2i). A152T neurons treated for 24 h with mTORi at the concentration for maximum tau reduction were stained for total tau or PHF1 P-tau (Fig. 2m, red) in combination with the neuronal marker MAP2 (Fig. 2m, green)

**Fig. 2 Compound dose-dependent reduction of tau in a neuronal cell model of tauopathy.** Tau-A152T (FTD19-L5-RC6) 6-week differentiated neurons were treated for 24 h, and tau protein levels and neuronal viability were measured. **a–d** Western blot of total tau (TAU5) and P-tau$^{S396}$, with representative blots shown on the left and mean densitometry shown on the right (**b–d** samples were run on the same gel, with the same vehicle sample, and image was cropped at the dotted line only for the purpose of this figure). Bands used for quantification are within brackets. Data points represent mean densitometry relative to vehicle-treated samples (a.u., arbitrary units) ± SEM of $n = 4$ biological replicates ($n = 6$ for vehicle and highest dose). **e–h** Tau ELISA dose-response curves. Data points represent mean tau per μg of total protein, relative to vehicle-treated samples ± SEM. Results represent $n = 4$ with technical replicates per ELISA plate. Statistical significance (**a–h**) was calculated using two-way ANOVA with post hoc Dunnett's multiple comparisons test relative to vehicle, with $^{ns}P > 0.05$, $*P \leq 0.05$, $**P \leq 0.01$, $***P \leq 0.001$, $****P \leq 0.0001$. **i–l** Dose-response curves for neuronal viability. Data points represent mean % viability relative to vehicle ± SEM, for $n = 3$ biological replicates. **m** IF of A152T neurons with total tau (K9JA) and P-tau$^{S396/S404}$ (PHF-1) antibodies in red, and neuronal marker MAP2 in green. Scale bars are 50μm. Representative images of $n = 3$. Source data are provided as a Source Data file.

and DNA (blue). Representative immunofluorescence (IF) images (Fig. 2m) show that mTORi did not affect MAP2 staining or cell density, while significantly reducing the levels of tau staining in cell bodies, with tau$^+$ neuronal processes still detected (Fig. 2m red).

We then tested mTORi on tau-P301L neurons and measured dose-effect on tau and neuronal viability. We treated 6-week differentiated neurons for 24 h with 1 nM–10 μM concentrations and used ELISA to measure total tau and P-tau$^{S396}$ (Supplementary Fig. 3a–d). The concentration at which maximum tau reduction was achieved for OSI-027, AZD2014, and AZD8055 was reduced to 5 μM in P301L neurons, with 60–70% down-regulation of tau (Supplementary Fig. 3b–d). Rapamycin was still the least potent compound with a maximum tau reduction of 50% (Supplementary Fig. 3a). Within the range of concentrations tested, the compounds did not affect P301L neuronal viability, except for rapamycin at ≥5 μM that caused a 40% loss of viability (Supplementary Fig. 3e). IF of mTORi-treated P301L neurons showed clear reduction in tau staining (Supplementary Fig. 3f, red) relative to the neuronal marker MAP2 (Supplementary Fig. 3f, green), with a less clear effect for rapamycin.

As a control experiment, we asked if mTORi affected tau in control neurons. Six-week differentiated tau-WT neurons were treated for 24 h at the established dose for maximum tau reduction in tauopathy neurons, and protein levels were measured (Supplementary Fig. 3g). We observed 50–60% down-regulation of total tau with AZD2014, AZD8055 and OSI-027, and <40% effect with rapamycin. In control neurons, the levels of P-tau$^{S396}$ and high MW species are relatively low (Supplementary Fig. 2a), and therefore the effect of compounds on P-tau was less pronounced (~40% reduction). Neuronal viability was not affected except for rapamycin at the highest dose (Supplementary Fig. 3h).

Reduction of tau by mTORi (10 μM, 3 μM rapamycin) was achieved without significant change in *MAPT* gene expression, as tested by qRT-PCR analysis of mTORi-treated A152T neurons, using probes that report on all *MAPT* isoforms (Supplementary Fig. 3i).

Next we asked whether mTORi affect tau preferentially based on protein solubility[21]. We treated A152T neurons with each compound for 24 h and performed protein fractionation based on Triton-X100 (T) and SDS (S) detergent solubility (Fig. 3a)[21,45,46]. Soluble forms of tau were isolated with the Triton buffer, whereas insoluble tau was extracted with 5% SDS. We found that OSI-027, AZD8055, and AZD2014 had a stronger effect reducing the levels of insoluble (S) tau by 60–90%, including oligomeric species (>250 kDa), relative to soluble (T) tau that showed an overall 50% decrease (Fig. 3b–d). Rapamycin had a similar effect on both soluble and insoluble tau and a weaker effect on high MW P-tau. In P301L neurons (Supplementary Fig. 4a–c) there was also a 50–80% downregulation of insoluble tau, with some variability in the relative change of total and P-tau$^{S396}$ relative to A152T neurons (Fig. 3d). This was expected, given that tau-A152T and

tau-P301L are predicted to originate different misfolded, oligomeric, and insoluble species, consistent with different brain pathologies, and therefore should have different propensities for clearance.

Altogether, our results show that, relative to rapamycin, AZD2014, AZD8055 and OSI-027 had a stronger effect on tau reduction, including insoluble protein, without loss of neuronal viability.

**Rescue of tau-mediated neuronal stress vulnerability.** In FTD patient-derived neurons, accumulation of tau and insoluble tau is coupled to increased neuronal vulnerability to stress[21,44]. Two of the stressors that revealed this phenotype were the highly aggregation-prone peptide Aβ(1-42), and high dose of the excitatory neurotransmitter NMDA. Both stressors promoted concentration-dependent loss of viability in A152T and P301L neurons, but not in non-mutant neurons (Supplementary Fig. 4d, e), revealing genotype-dependent vulnerability. This phenotype was rescued by *MAPT*-Kd in A152T neurons, demonstrating tau-dependent toxicity[21]. Here, we asked whether mTORi, by reducing tau levels, were protective against stress vulnerability.

A152T neurons were pre-treated with vehicle (DMSO) or mTORi for 8 h at the concentration leading to >70% tau reduction (10 μM, 3 μM rapamycin), followed by addition of the stressor, 30 μM Aβ(1-42) or 400 μM NMDA, for 16 h. At 24 h, neuronal viability was measured (Supplementary Fig. 4f-ii). Vehicle- and mTORi-alone treated samples served as controls (Supplementary Fig. 4f–i). Vehicle-only pre-treated neurons stressed with Aβ(1-42) showed 70% loss of viability, that is, viability was 30% of vehicle-control (Fig. 3e–h). Pre-treatment with rapamycin rescued viability to 60% of vehicle-control (Fig. 3e), OSI-027 rescued viability to 80% of vehicle-control (Fig. 3f), AZD2014 rescued viability to 100%, phenocopying *MAPT*-Kd neurons (Fig. 3g), and AZD8055 rescued viability to 70% of vehicle-control (Fig. 3h). As expected, 24h-treatment with each mTORi alone did not affect viability (Fig. 3e–l). Next, we measured the rescue of viability loss caused by NMDA. Vehicle-only pre-treated neurons stressed with NMDA showed a 60% loss of viability, that is, viability was 40% of vehicle control (Fig. 3i–l). Pre-treatment with rapamycin showed the weakest effect by rescuing viability to only 60% of vehicle-control (20% improvement, Fig. 3i), whereas OSI-027 rescued viability to 80% (Fig. 3j), and both AZD2014 and AZD8055 showed the strongest effect by rescuing viability to ~90% of vehicle-control, closely phenocopying *MAPT*-Kd (Fig. 3k, l).

We also tested the protective effect of mTORi in P301L neurons and observed a similar trend (Supplementary Fig. 4g–n). Both Aβ(1-42) and NMDA caused 50–60% loss of viability (i.e., 40–50% of vehicle alone). Pre-treatment with rapamycin had a modest effect and rescued viability to ~70% of vehicle control (Supplementary Fig. 4g, k), representing a 20% improvement. OSI-027, AZD2014 and AZD8055 showed, in general, a rescue in viability corresponding to 80–100% of

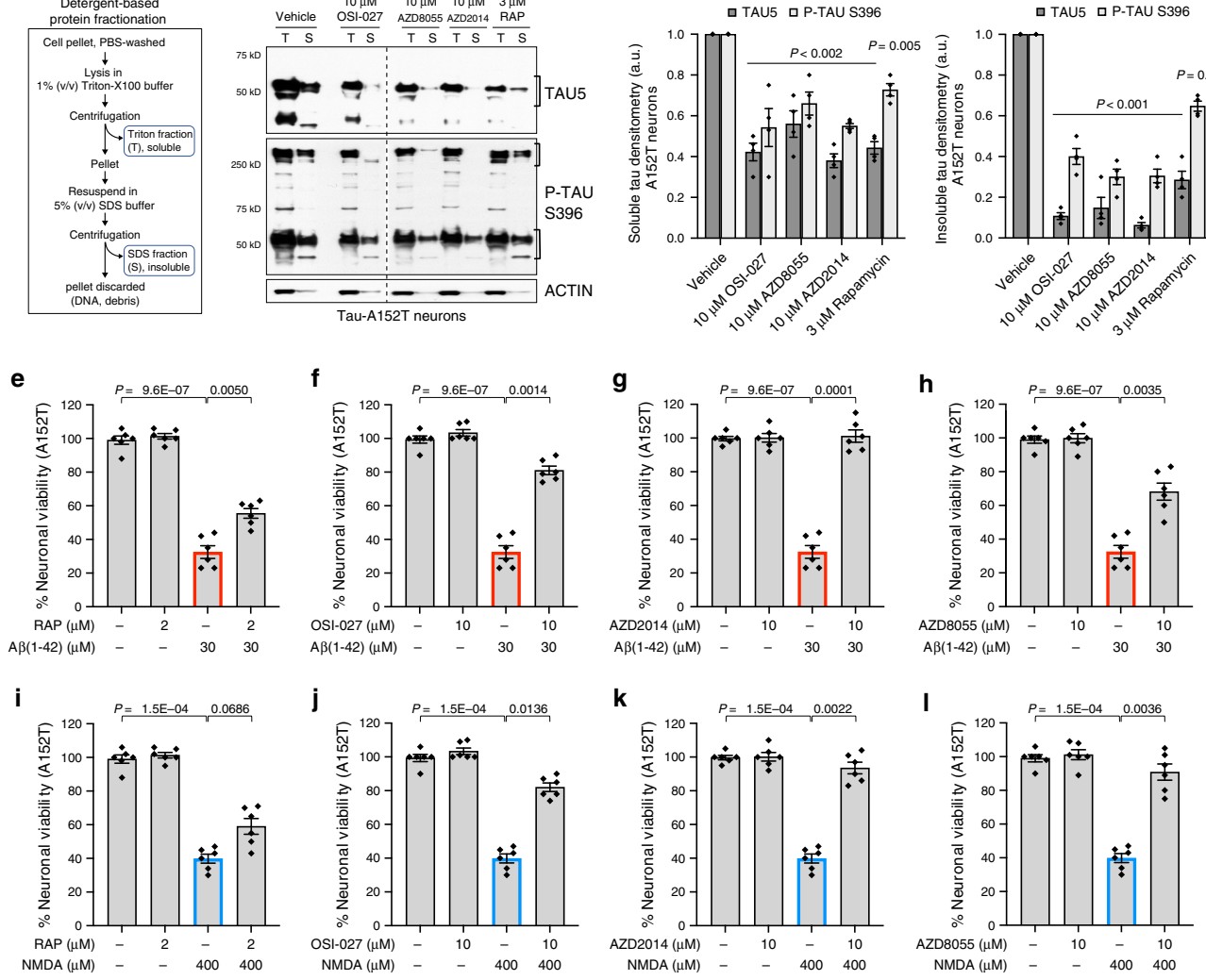

**Fig. 3 Rescue of tau phenotypes in 8-week differentiated A152T (FTD19-L5-RC6) neurons upon 24 h treatment. a** Summary of the assay for protein fractionation based on Triton-X (T) and SDS (S) detergents differential solubility. **b–d** Western blot analysis of tau after mTORi compound treatment and detergent fractionation. **b** Samples were run on the same blot, image cropped (dotted line) only for the purpose of this figure. Brackets indicate protein bands for densitometry analysis. Graph bars represent mean densitometry ± SEM, and black diamond-dots indicate individual data points for soluble (**c**) and insoluble (**d**) total tau (TAU5) and P-tau$^{S396}$ levels relative to vehicle samples. $n = 4$ biological replicates. Statistical significance relative to vehicle was calculated using two-tailed unpaired $t$-test and $P$ values are indicated in each graph. **e–l** Rescue of neuronal vulnerability to 30 μM Aβ(1-42) (red) and 400 μM NMDA (blue), by pre-treatment with rapamycin (**e, i**), OSI-027 (**f, j**), AZD2014 (**g, k**) or AZD8055 (**h, l**). Assay summary in Supplementary Fig. 4f. Bars represent mean % viability relative to vehicle-treated neurons ± SEM, and black diamond-dots represent individual data points for $n = 3$ biological replicates with 2 technical replicates per experiment. Statistical significance was calculated with a two-tailed unpaired $t$-test and $P$ values are indicated ($^{ns}P > 0.05$, $^{*}P \leq 0.05$, $^{**}P \leq 0.01$, $^{***}P \leq 0.001$, $^{****}P \leq 0.0001$). Source data are provided as a Source Data file.

vehicle-control (Supplementary Fig. 4h–j, m, n). The only exception was OSI-027+NMDA that showed a non-significant <10% improvement (Supplementary Fig. 4l).

In conclusion, concomitant with tau reduction, mTORi rescued vulnerability to stress in tauopathy neurons, with higher efficacy for mTORC1/2 inhibitors relative to rapamycin. Under the hypothesis that reduction of tau levels and toxicity occurs downstream of mTOR inhibition and autophagy activation, we investigated the mechanism-of-action to dissect compound primary (mTOR) and secondary (autophagy) targets (Supplementary Fig. 1b).

**Evidence of autophagy activation in human ex vivo neurons.** To investigate whether tau reduction by mTORi was caused by autophagy clearance, we monitored the ALP by measuring changes in specific markers (Fig. 4a)[47]. Since it is not known how

much tau downregulation is sufficient for therapeutic effect, we treated A152T neurons under the conditions leading to maximum tau reduction (70–80%), and measured ALP changes at 24 h.

Upon mTORC1 inhibition, and via changes in phosphorylation of ULK1, VPS34 and Beclin 1 (BECN1), LC3-I is converted into LC3-II during autophagosome formation, and ATG12 is covalently bound to ATG5 and targeted to autophagosomes[48] (Fig. 4a). Upon mTORi treatment, we observed an upregulation of LC3-II (threefold) and ATG12/ATG5 (>2-fold), relative to vehicle-treated neurons (Fig. 4b–d), consistent with autophagosome formation. Then, LC3-II interacts with p62 that transports cargo into the autophagosomes and is itself degraded[47]. Phosphorylation of p62 at S403, is essential for its function to sequester cargo into autophagosomes[23,49]. In our study, mTORi caused a 50% reduction in p62 levels, and >50-fold upregulation

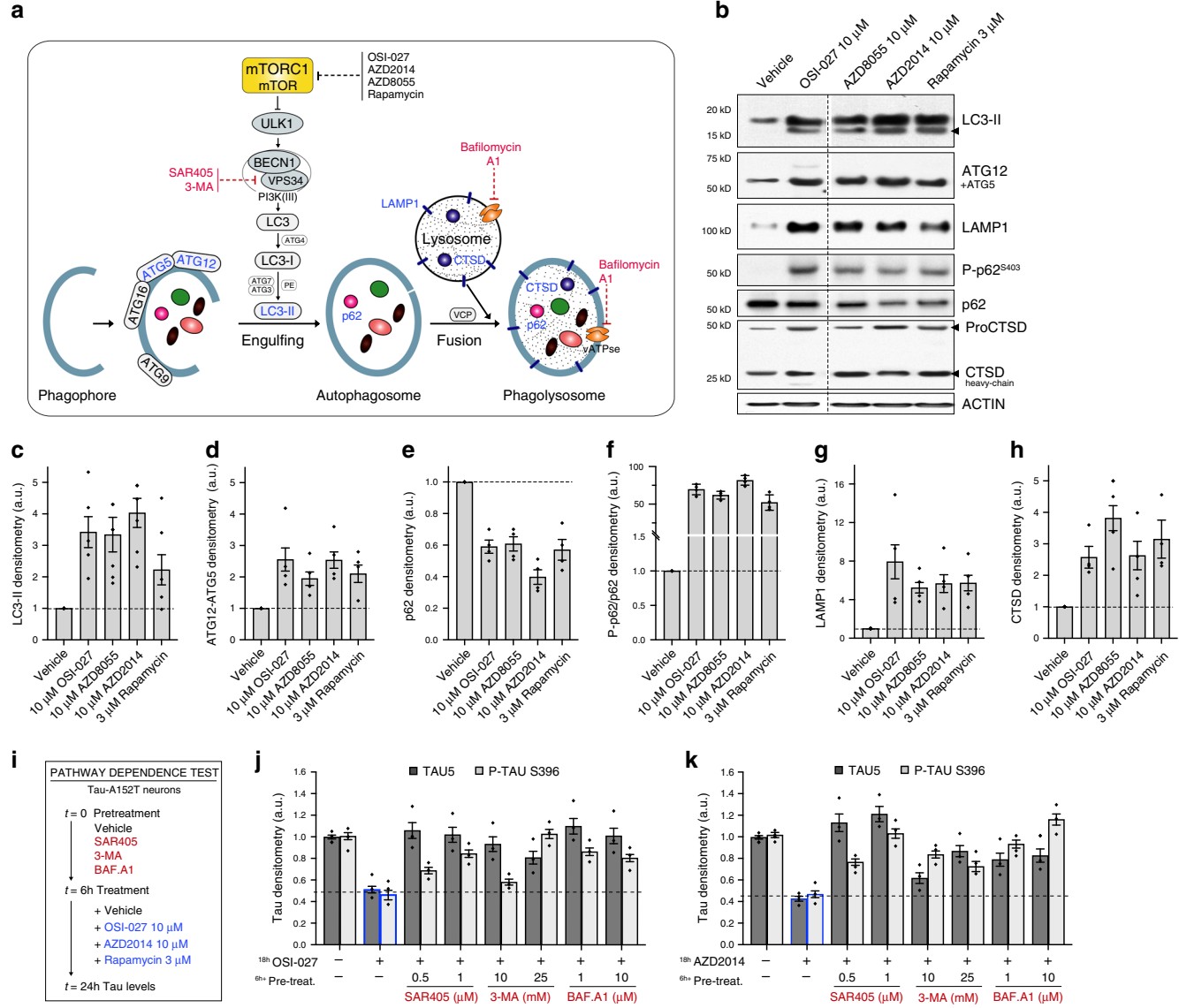

**Fig. 4 Compound activation of autophagy as shown by upregulation of pathway-specific markers and pathway-dependence analysis. a** Simplified schematic of the proposed mechanism for mTORi-mediated tau clearance through autophagy. mTORi are shown in black, and autophagy inhibitors in red. **b** Western blot of autophagy-specific markers (in blue, **a**) in A152T neurons (6-week differentiated, FTD19-L5-RC6) upon 24 h treatment (samples were run on the same gel, image was cropped on the dotted line only for the purpose of this figure). Blot is representative of $n = 4$ biological replicates. **c–h** Graph bars represent mean densitometry relative to vehicle ± SEM, and black diamond-dots indicate individual data points for $n = 4$ biological replicates ($n = 5$ for LC3-II). **i** Assay to test mTORi effect pathway-dependence: A152T (FTD19-L5-RC6) neurons were treated for 6 h with autophagy inhibitors (SAR405, 3-MA, BAF.A1) followed by mTORi (10 μM OSI-027, 10 μM AZD2014) for a total of 24 h. **j, k** Western blot densitometry of total tau (TAU5) and P-tau$^{S396}$ levels. Graph bars represent mean tau densitometry ± SEM and black diamond-dots indicate individual data points for $n = 3$ biological replicates. Corresponding representative blots are included in Supplementary Fig. 5b, c. Source data are provided as a Source Data file.

of P-p62$^{S403}$ relative to total p62 and vehicle-control (Fig. 4b, e, f). This trend is consistent with enhanced autophagy flux, even though the fold-change in P-p62$^{S403}$ is inflated by normalization to vehicle-treated samples where P-p62$^{S403}$ is mostly undetected (Fig. 4b). Lysosomal LAMP1[50] was also upregulated by >4-fold upon treatment (Fig. 4b, g). Finally, cathepsin D (CTSD), a lysosomal aspartyl protease (Fig. 4a) that forms a 46 kDa intermediate active proCTSD, is cleaved in the lysosome to produce a 28 kDa heavy-chain and a 15 kDa light-chain active subunits[51]. To examine CTSD associated with lysosomal activity, we measured the levels of the 28 kDa heavy-chain CTSD (Fig. 4b, h), which was upregulated by ≥2.5-fold by each mTORi. Upregulation of autophagy components was also observed at the gene expression level, with >2-fold increase in *LAMP1*, *CTSD*,

and *p62* mRNA levels, without change in *MAPT* expression (Supplementary Fig. 5a).

To further corroborate that the effect of mTORi on tau was a consequence of autophagy, we prevented autophagy initiation or clearance pharmacologically by pre-treating A152T neurons for 6 h with 3-MA[52], SAR405[53], bafilomycin A1 (BAF.A1)[54] or vehicle (DMSO) alone, followed by addition of mTORi for 18 h (Fig. 4i). Both 3-MA and the more selective SAR405 inhibit autophago-some formation through VPS34 inhibition[55] (Fig. 4a). In contrast, BAF.A1 is a H$^+$-vATPase inhibitor that prevents lysosome-autophagosome fusion, inhibiting proteolysis (Fig. 4a)[54]. We tested whether these compounds would negate the effects of OSI-027, AZD2014, or rapamycin (Fig. 4j, k, Supplementary Fig. 5b–e). Neurons pre-treated with vehicle-alone and then treated for 18 h

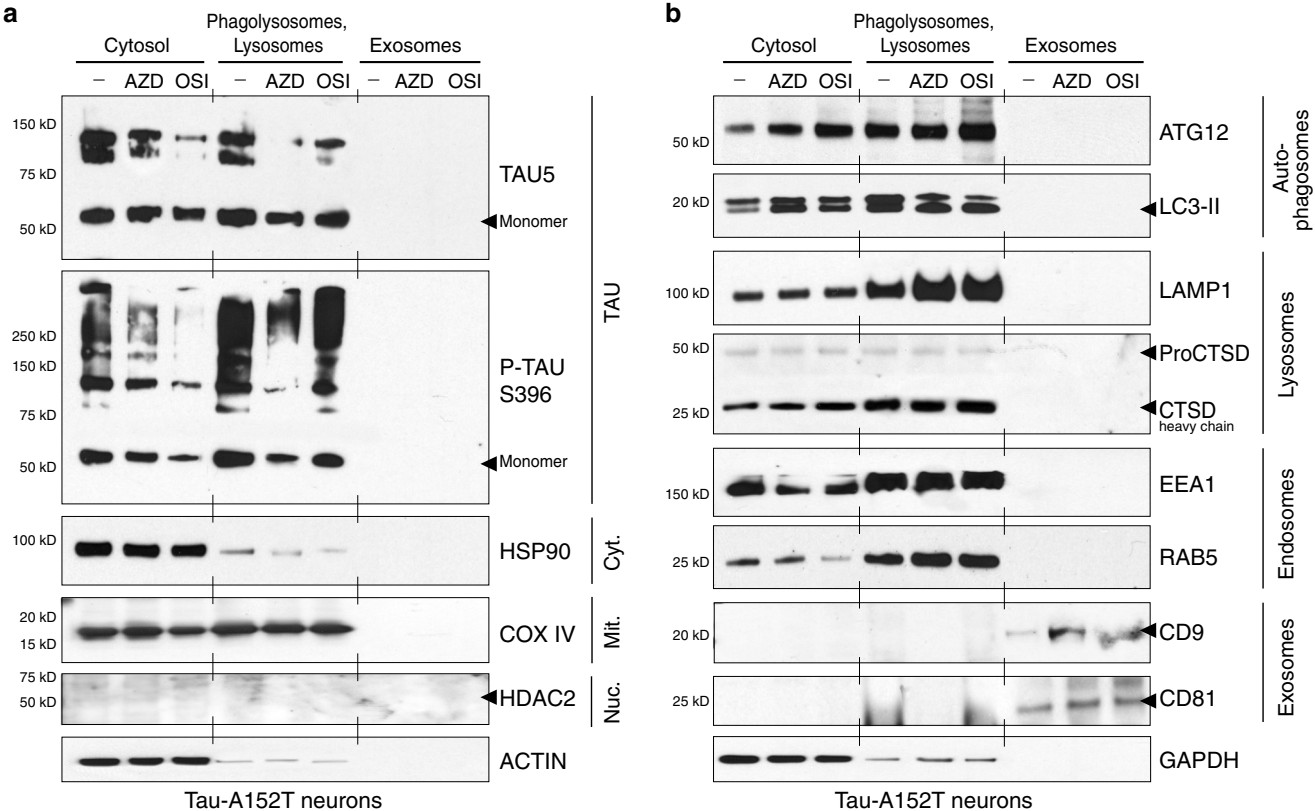

**Fig. 5 mTORi promoted tau sequestration into phagolysosomes.** Tau-A152T neurons (6-week differentiated FTD19-L5-RC6) were treated with vehicle (−), 10 μM AZD2014 (AZD) or 10 μM OSI-027 (OSI) for 8 h. Cell lysates were fractionated by density gradient and western blot analysis shows **a** total tau (TAU5) and P-tau$^{S396}$ levels in cytosol, lysosomes/phagolysosomes and exosomes fractions; and **b** fraction-specific markers. Key: Nuc, nuclear negative control; Cyt, cytosol; Mit, mitochondrial. Blots shown are representative of $n = 2$ biological replicates.

with 10 μM OSI-027 (Fig. 4j) or 10 μM AZD2014 (Fig. 4k) showed a 60% downregulation in total tau and P-tau$^{S396}$. However, when pre-incubated with SAR405, 3-MA or BAF.A1, there was a dose-dependent suppression of the effect of each mTORi, with the highest doses maintaining tau levels at 90–100% of vehicle-treated neurons (Fig. 4j, k). This means that although mTOR inhibition by OSI-027 and AZD2014 occurs upstream (Fig. 4a), SAR405, 3-MA, and BAF.A1 disrupt autophagy flux and clearance of tau. Because the effect on tau by rapamycin was more moderate, it was reversed at lower concentrations of autophagy inhibitors (Supplementary Fig. 5d, e).

Altogether, these results demonstrated that mTORi promoted clearance of tau in human neurons by autophagy. While static analysis of individual markers is not indicative of autophagy flux, the overall effects observed on multiple markers and in combination with autophagy inhibitors are consistent with increased autophagy flux by mTORi. Importantly, clearance of tau was observed in a disease context, in FTD patient-derived neurons, where the ALP is already affected, revealing that this pathway can be pharmacologically enhanced to rescue tau pathogenicity and possibly neurodegeneration.

**Tau is recruited into lysosomes of mTORi-treated neurons.** We sought to establish that mTORi promoted tau sequestration into autophagosomes and clearance in phagolysosomes (Fig. 4a), excluding an alternative mechanism where tau would be incorporated into exosomes and secreted. Exosomes carrying misfolded proteins such as tau[56] can be neuroprotective, by clearing toxic proteins in their lumen[57], but can also enhance propagation of neurodegeneration[58,59].

Neurons expressing tau-A152T (Fig. 5) or tau-P301L (Supplementary Fig. 6) were treated with vehicle, 10 μM AZD2014 or 10 μM OSI-027 for only 8 h to capture tau *en route* within the ALP, before maximum degradation. Neuronal lysates were processed through density gradient ultracentrifugation for subcellular fractionation and cell media was collected for exosomes purification. All fractions were analyzed by western blot of tau and fraction-specific markers (Fig. 5, Supplementary Fig. 6). Under basal conditions (vehicle-treated, −) A152T neurons showed a high proportion of tau in the lysosomes, corroborating that tau is a physiological substrate of the ALP, without tau detection in exosomes (Fig. 5a). After mTORi treatment, tau and P-tau$^{S396}$ were found enriched in lysosomes/phagolysosomes, relative to the cytosol. After 8 h, there was already some clearance of tau, as shown by reduction in tau with AZD2014 (AZD) and OSI-027 (OSI) in the cytosolic fraction, relative to vehicle (−), a difference also observed in the lysosome fraction (Fig. 5a). After mTORi treatment, under the sensitivity limits of this assay, tau was also not detected in exosomes. Cytosolic (HSP90), mitochondrial (COX IV) and nuclear (HDAC2) markers were used as controls to ensure that the appropriate fractions were analyzed (Fig. 5a). As such, the chaperone HSP90 was detected mainly in the cytosol, the mitochondrial cytochrome c oxidase (COX IV) was detected in the cytosol and lysosome fractions, indicating that this organelle co-purified with lysosomes/phagolysosomes, and HDAC2 was not detected, indicating that the nuclear fraction was successfully excluded. Autophagosomal (ATG12, LC3-II) and lysosomal (LAMP1, CTSD) markers were substantially enriched in the purified lysosomes/phagolysosomes (Fig. 5b), validating the interpretation of results for tau. We also confirmed that lysosome precursor and endosome-specific markers such as EEA1 (early

endosome antigen 1) and the endosome sorting Ras-related protein RAB5, were enriched in the lysosome fraction (Fig. 5b). Finally, we verified presence of exosomes in cell media by detecting exosome membrane proteins CD9 and CD81[58,59] in the respective fraction, with a slight increase in band intensity for mTORi-treated samples (Fig. 5b).

Next, we evaluated tau distribution in sub-cellular fractions of P301L neurons, under the same conditions (Supplementary Fig. 6). The most crucial difference was tau detection in exosomes of vehicle-treated (–) but not in mTORi-treated samples, under the limits of detection of this assay. Exosomal tau has also been reported in a P301L transgenic mouse model[60]. But, upon AZD2014 or OSI-027 treatment, the majority of tau was found in phagolysosomes (Supplementary Fig. 6a). Another difference in P301L neurons was that the intensity of total tau and P-tau$^{S396}$ in lysosomes was much higher relative to the cytosol (Supplementary Fig. 6a). This might be due to strong tau recruitment into phagolysosomes and/or higher autophagic activity in these neurons, as supported by higher levels of autophagosome (LC3-II), lysosome (LAMP1) and endosome (EEA1) markers (Supplementary Fig. 6b), compared with A152T neurons (Fig. 5b).

In these experiments, we detected tau species of high MW in A152T neurons (~100 kDa, Fig. 5a) and in P301L neurons (≥100–250 kDa, Supplementary Fig. 6a) with the total tau (TAU5) antibody, which were not detected with higher % SDS lysis buffer (Fig. 2a–d). The mildly denaturing conditions employed here were specific for lysosomes isolation without disruption of organelle membrane and content, possibly leading to more incomplete oligomeric-tau dissociation, particularly enriched in P301L neurons.

Taken together, the results corroborate that, upon mTORi treatment of neurons, tau was recruited into autophagosomes and phagolysosomes for autophagy-mediated degradation.

**Tau clearance is a downstream effect of mTOR inhibition.** Little is known about the mechanism-of-action of mTORi in human neurons or in a tauopathy context. Rapamycin is a known allosteric inhibitor of mTORC1[32,61], whereas OSI-027, AZD8055, and AZD2014 have been shown, in vitro and in cancer cells, to be dual mTORC1/2 catalytic-site inhibitors (Fig. 6a)[32,34–36]. The complex mTORC1 directly phosphorylates p70S6K on T389, which phosphorylates the ribosomal protein S6 on S240/244, and 4EBP1 on T37/46[62]. The mTORC2 complex phosphorylates AKT on S473[32,34,61]. Also, mTOR is negatively regulated by the energy sensor AMP-activated protein kinase (AMPK)[63] that we sought to exclude as an indirect target (Fig. 6a).

To corroborate OSI-027, AZD2014, and AZD8055 inhibition of the serine/threonine mTOR kinase activity, we measured specific substrates phosphorylation (Fig. 6a). To assess effect independent of disease context, we treated tau-WT neurons for 24 h with OSI-027, AZD2014 and AZD8055 (Fig. 6b–d, Supplementary Fig. 7a) at concentrations between 0.1 nM and 50 μM, which did not cause noticeable toxicity. We observed a dose-dependent downregulation of P-p70S6K$^{T389}$ and P-4E-BP1$^{T37/46}$, which was a result of mTORC1 decreased activity, whereas P-S6$^{S240/244}$ was downregulated due to reduced p70S6K activity upon mTOR inhibition. Maximum mTOR inhibition was achieved between 1 μM and 10 μM doses, with 100% loss of P-4E-BP1$^{T37/46}$ and 70–100% reduction in P-p70S6K$^{T389}$. The p70S6K substrate S6$^{S240/244}$ also showed a reduction of 80–90% at >1 μM (Fig. 6b–d). These results reveal that our compounds targeted mTORC1 in neurons, and that maximum mTORC1 inhibition occurred approximately at the same doses leading to greatest tau clearance.

To measure compound effect on mTORC1 activity in tauopathy A152T neurons, we treated cells for 24 h and measured the levels of mTORC1 substrates phosphorylation (Fig. 6e), including P-mTOR$^{S2448}$, which can result from mTOR auto-phosphorylation or p70S6K activity feedback-loop[64] (Fig. 6a). At 10 μM (3 μM rapamycin) the strongest reduction observed was in P-mTOR$^{S2448}$ (70–80%) and P-S6$^{S240/244}$ (80–90% lower) (Fig. 6f, h), which implicates reduced mTOR kinase activity and also reduced p70S6K activity. P-p70S6K$^{T389}$ showed a more modest reduction of ~50% (Fig. 6g), which might have to do with relative antibody immunoreactivity sensitivity or different target-engagement and inhibition kinetics in WT (Fig. 6b–d) vs. A152T neurons (Fig. 6g). Furthermore, mTOR negatively regulates autophagy through direct phosphorylation of the autophagy-initiating serine/threonine kinase ULK-1 (Fig. 6a), which in turn phosphorylates and activates BECN1 (P-BECN1$^{S15}$) within the nucleation complex for autophagosome formation (Figs. 4a and 6a). Upon treatment of A152T neurons, we detected an upregulation of P-BECN1$^{S15}$ relative to total BECN1 (Fig. 6e, i), which is consistent with mTOR inhibition, and subsequent ULK-1 and autophagy activation.

Compound effect on mTORC2 activity was assessed by measuring the levels of P-AKT$^{S473}$. In A152T neurons treated with 10 μM mTORi (3 μM rapamycin) for 24 h, we did not observe significant effect on P-AKT$^{S473}$ relative to total AKT (Supplementary Fig. 7b, c). This suggests that in neurons, under these experimental conditions, OSI-027, AZD8055, and AZD2014 primarily inhibited mTORC1. This is in contrast with data from cancer cell and mouse models, where these are dual mTORC1/2 inhibitors that downregulate P-AKT$^{S473}$. Lastly, treatment of A152T neurons did not cause measurable changes in P-AMPKα$^{T172}$ (Supplementary Fig. 7b, d), again pointing to a mTORC1-mediated effect on autophagy and tau.

To further demonstrate that mTOR inhibition drives autophagy clearance of tau, we co-treated A152T neurons with each mTORi and a potent mTOR activator, MHY-1485 (Fig. 6a, j, k)[65]. When neurons were pre-treated with MHY-1485 for 6 h (1 μM, 2.5 μM) and then rapamycin, OSI-027 or AZD2014 (Fig. 6j), tau levels were restored to ~100% (Fig. 6k), relative to the >60% tau reduction observed with mTORi alone (18 h, Fig. 6k, blue-line bars). These results corroborated that clearance of tau by each mTORi was dependent on mTOR inhibition.

**mTORi have a prolonged effect on tau levels and toxicity.** So far, we examined the mechanism-of-action of mTORi after 24h-treatment of neurons. Now, we asked what the effect on tau protein beyond this time-point was. We considered three possible scenarios: (a) tau quickly returns to basal levels after 24 h, (b) tau levels continue to decrease followed by a slow recovery to basal levels, or (c) tau recovers beyond basal levels due to changes in kinetics of protein accumulation and clearance. To investigate this, we focused on OSI-027 and AZD2014 treatment of A152T neurons, using rapamycin as a control. We treated 6-week differentiated neurons with 10 μM mTORi (3 μM rapamycin) for 24 h, washed the cells and added new culture media without compound (Fig. 7a). Neurons were collected over a time-course between 1 and 20 days, and tau levels and neuronal viability were measured. Day 0 corresponds to vehicle-treated samples. After 24 h treatment (day 1), we observed the expected 60% reduction in tau levels by OSI-027 (Fig. 7c) and AZD2014 (Fig. 7d), and 40% reduction by rapamycin (Fig. 7a). Surprisingly, after media washout, the levels of total tau and P-Tau$^{S396}$ continued to decrease. For OSI-027, by day 8 tau was down 80% (20% of vehicle-treated neurons, Fig. 7c), whereas AZD2014 maximum effect was seen at day 12 with 80–90% reduction in tau (Fig. 7d).

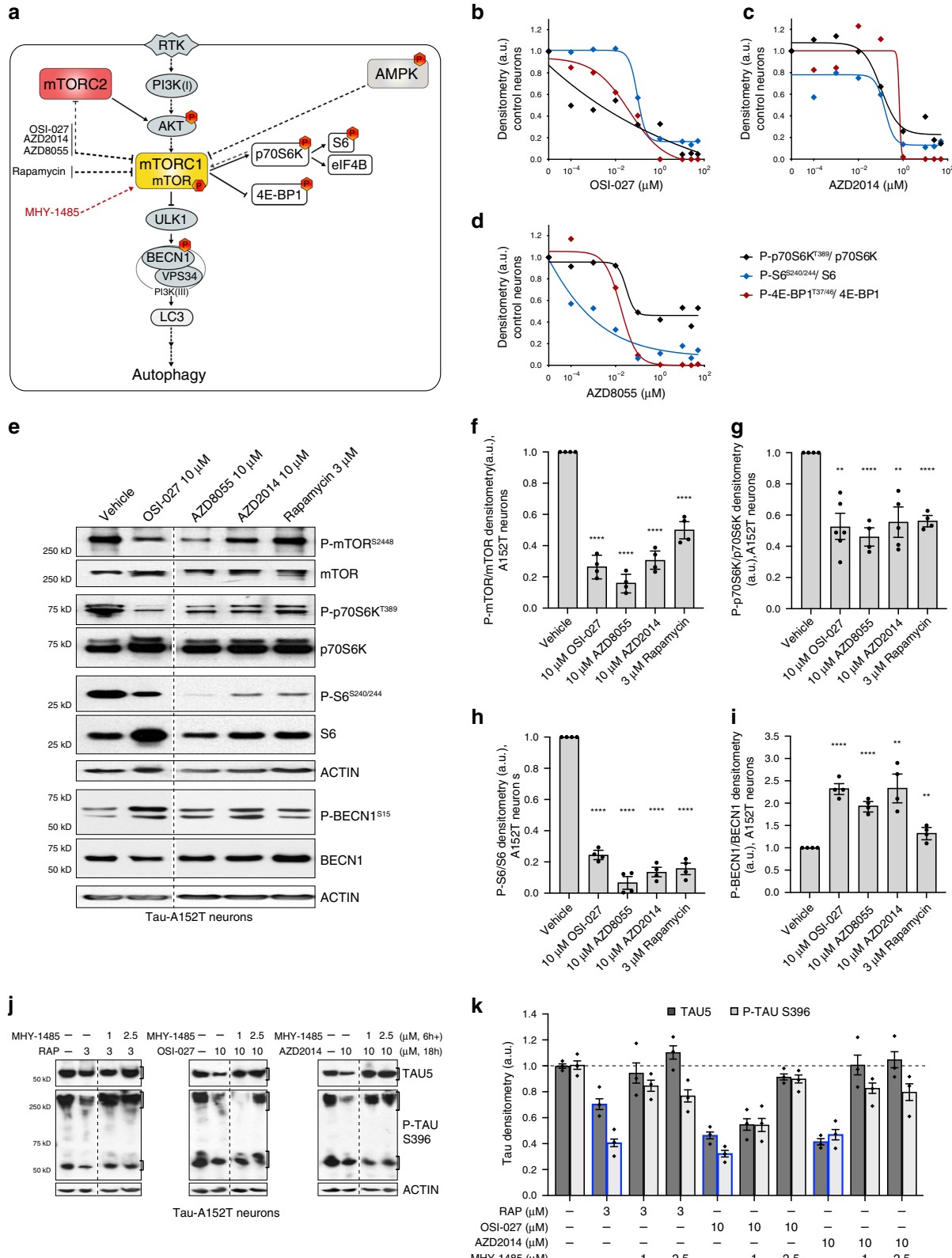

Even for rapamycin, the levels of tau still dropped further until day 4 (Fig. 7a). After day 12 for rapamycin, but only after day 16 for OSI-027 and AZD2014, protein levels started to recover, and by day 20 either total tau, P-tau[S396] or both were back to basal levels (Fig. 7b–d). We repeated this experiment with 100 nM OSI-027 (Fig. 7e) and 100 nM AZD2014 (Fig. 7f), to test whether the prolonged effect on tau would be observed at the EC$_{50}$. We

confirmed ~50% clearance of tau at 24 h, and the levels of total tau and P-tau[S396] continued to decrease until day 12 for OSI-027 (60–80% reduction from vehicle-alone) and until day 4 for AZD2014 (60–80% reduction from vehicle-alone). The levels of tau were not as reduced and did not persist low as long as with 10 μM treatment, with a faster recovery to basal levels (compare Fig. 7c vs. 7e, Fig. 7d vs. 7f). Potentially, higher mTORi

**Fig. 6 Demonstration of mTORC1 as the primary target of mTORi in human *ex vivo* neurons. a** Simplified schematic of the targeted pathways by mTORi (in black), through mTORC1 and autophagy clearance of tau. Red-hexagons indicate phospho-substrates measured. Dotted lines indicate predicted or indirect interactions. **b–d** Compound dose-response curves (0.1 nM–50 μM) for mTORC1 activity in Control-1 neurons (6-week differentiated 8330-8-RC1, 24 h treatment), measured by western blot of substrates phosphorylation (representative blots in Supplementary Fig. 7a). Data points represent mean densitometry of each phospho-marker normalized to total levels, for $n = 2$ biological replicates. **e–i** Western blot of A152T neurons (6-week differentiated FTD19-L5-RC6) treated for 24 h and immunoprobed for mTORC1 substrates and BECN1 levels and phosphorylation. Representative blot (**e**) of $n = 2$ biological replicates and 2 technical replicates. Graph bars (**f–i**) represent mean densitometry of phospho-markers normalized to total levels of each protein, relative to vehicle ± SD. Black circles represent individual data points for $n = 2$ biological replicates and 2 technical replicates. Statistical significance was calculated with a two-tailed unpaired $t$-test (\*\*$P \leq 0.01$, \*\*\*\*$P \leq 0.0001$). **j**, **k** A152T neurons (FTD19-L5-RC6) were pre treated with mTOR activator MHY-1485 (**a**) for 6 h, followed by rapamycin, OSI-027 or AZD2014 for a total of 24 h. Effect on total tau (TAU5) and P-tau$^{S396}$ was measured by western blot (representative images of $n = 3$), relative to the effect of each mTORi alone (bars outlined in blue, **k**). Graph bars represent mean tau densitometry ± SEM and diamond-dots represent individual data points for $n = 3$ biological replicates. Samples (**e**, **j**) were run on the same gel and images were cropped at the dotted lines only for the purpose of this figure. Source data are provided as a Source Data file.

concentration led to slower tau recovery due to longer target occupancy and sustained autophagy activity.

A parallel time-course analysis of neuronal viability showed that 10 μM OSI-027 had no significant effect on viability up to day 16 (Fig. 7g). Strikingly, 10 μM AZD2014 caused an increase of neuronal viability by 25% until day 4, which then returned to 100% of vehicle-treated control (Fig. 7h). Rapamycin (3 μM) did not affect viability for about 2 days, but then there was a progressive loss of viability (Fig. 7i). To determine whether tau reduction had a more subtle effect on neuronal integrity and neurodevelopment, such as by restricting neurite outgrowth, we repeated the time-course analysis after mTORi treatment and assessed the levels of the microtubule protein β-III-tubulin (Fig. 7b–f). Within the first 24–48 h post treatment there was a trend for ~10–20% reduction in β-III-tubulin with the highest concentrations of mTORi (Fig. 7b–d), but by day 4 onward levels were back at 100%. Given that reduction of tau levels persisted for a longer period of ≥12 days, this suggests that tau reduction and effect on β-III-tubulin were independent, and that neuronal integrity was not affected as per β-III-tubulin and actin levels over time.

To establish that prolonged mTORi effect on tau was not specific to A152T neurons, we repeated this experiment (Fig. 7a) in P301L neurons and WT neurons (Supplementary Fig. 8). The initial observations were corroborated with a few key differences. In P301L neurons, both OSI-027 and AZD2014 promoted an initial increase of cellular viability (25–50%) that overlapped with the timeframe of tau reduction between 1 and 12 days (Supplementary Fig. 8b, c). By day 20 the levels of tau had not yet recovered to 100%, suggesting that perhaps tau-P301L takes longer to (re-)accumulate in this neuronal system. In WT neurons, both OSI-027 and AZD2014 had a strong effect on tau clearance between 1 and 4 days (70–90% reduction relative to vehicle-alone, Supplementary Fig. 8e, f). Concomitant with tau reduction, OSI-027 increased neuronal viability by ~30% (Supplementary Fig. 8e), whereas AZD2014 did not significantly affect viability of control neurons (Supplementary Fig. 8f). Rapamycin showed variable effects across cell lines and biological replicates (Supplementary Fig. 8a, d), but promoted significant tau clearance and improved viability in WT neurons. Finally, the levels of β-III-tubulin were not significantly affected in WT or P301L neurons, except for an ~10% reduction within the initial day of treatment with OSI-027 and AZD2014, followed by rapid recovery (Supplementary Fig. 8b, c, e, f). The effect by rapamycin was again variable with 20–30% reduction in β-III-tubulin that persisted through the 20 days of analysis (Supplementary Fig. 8a, d), suggesting that rapamycin could have detrimental effects on neurodevelopment.

Altogether these results demonstrate that mTORi promoted tau clearance at 24 h and this effect persisted for 12–16 days,

without reducing neuronal viability or integrity, and in fact promoting an increase in viability for about 4 days post treatment. This new and unexpected observation, validated in three independent and distinct cellular models, supports the therapeutic potential of mTORi for tauopathies.

To address the biological relevance of these findings, we examined if the prolonged effect on tau would affect toxicity as measured by vulnerability to the stressors Aβ(1-42) and NMDA (Supplementary Fig. 4d, e). We treated A152T neurons with 10 μM OSI-027, AZD2014 or vehicle for 24 h followed by compound-media washout. We then performed stress assays at day 1 (24 h control), and at days 2, 6, 8 and 16, by challenging neurons with either Aβ(1-42) or NMDA for 18 h, followed by viability measurement (Fig. 8a). Over this time-course, Aβ(1-42) and NMDA caused a 50–70% loss of viability in vehicle pre-treated neurons (Fig. 8b). However, in OSI-027 and AZD2014 pre-treated neurons, we observed a protective effect with much weaker viability loss by Aβ(1-42) or NMDA. Notably, this protective effect was stronger on the days corresponding to lower levels of tau (days 2–6), and by day 16 was practically lost (Fig. 8b).

**Prolonged impact of mTORi on mTORC1 activity and autophagy.** Given the prolonged effect of mTORi on tau levels and toxicity after a single dose, we examined if this was associated with a sustained effect on autophagy activity. Following the same setup (Fig. 7a), we treated A152T neurons with rapamycin, OSI-027 and AZD2014, and for a period of 20 days measured the levels of LC3-II, ATG12/5, LAMP1 and p62 (Fig. 8c–f, h–k, m–q, Supplementary Fig. 9a). Results showed an upregulation of LC3-II (Fig. 8c, h, m), ATG12 (Fig. 8d, i, n) and LAMP1 (Fig. 8e, j, o) upon the 24h-treatment that persisted, in general, above vehicle-treated levels until day 4–8, followed by a progressive return to basal levels. Exceptions included LC3-II levels that remained upregulated by 8–10-fold by day 20 (Fig. 8c, h, m), and LAMP1 levels in AZD2014-treated neurons that remained threefold upregulated by day 20 (Fig. 8o). Conversely, p62 levels were downregulated for at least 8 days post treatment, followed by recovery to vehicle-treated levels (Fig. 8f, k, p). We conclude that mTORi had a prolonged effect on tau reduction in concert with sustained autophagy activity.

To complement these findings, we tested the levels of mTORC1 substrates' phosphorylation to assess mTOR activity for the time-course of 20 days post treatment (Figs. 6a and 7a), and measured P-p70S6K$^{T389}$, P-S6$^{S240/244}$, and P-4E-BP1$^{T37/46}$, relative to total levels of each substrate (Fig. 8g, l, q, Supplementary Fig. 9b). We observed a similar time-dependent reduction in mTORC1 substrates phosphorylation relative to autophagy activity and tau levels, i.e., a steep decrease within

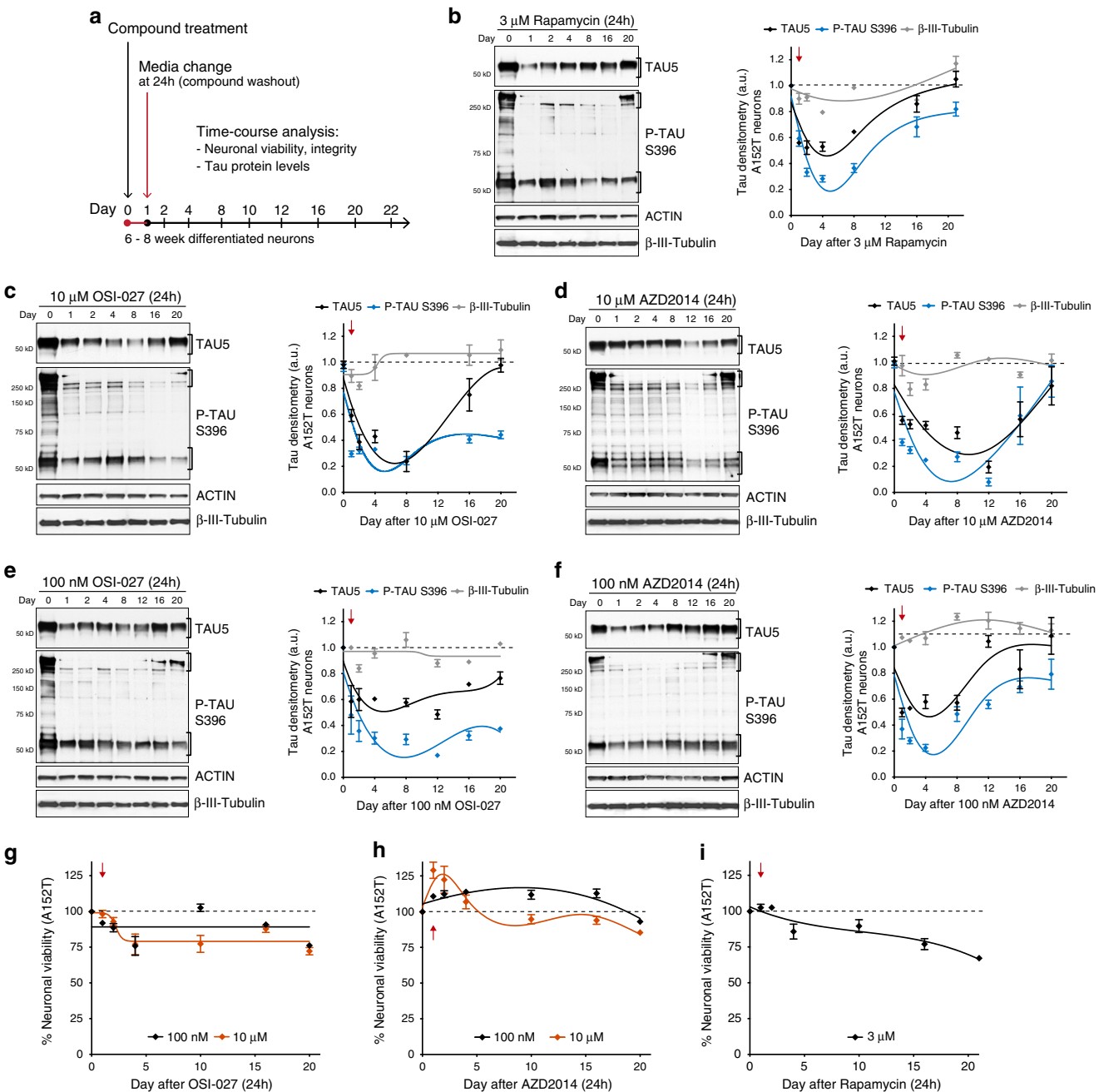

**Fig. 7 Single-dose mTORi has sustained effect on tau phenotypes. a** Assay to measure effect of mTORi 24h-treatment on tau levels and neuronal viability over a time-course of 20 days. **b–i** A152T neurons (6-week differentiated FTD19-L5-RC6) were treated with rapamycin, OSI-027 or AZD2014 for 24 h, followed by compound washout (red arrow). **b–f** Total tau (TAU5), P-tau$^{S396}$ and the neuronal microtubule marker β-III-tubulin were measured by western blot over a period of 20 days post treatment. Representative blots are shown, and graph data points represent mean protein densitometry (bands within brackets) ± SEM for $n = 3$ biological replicates. Dotted lines correspond to vehicle-treated protein levels. **g–i** Time-course analysis of effect on neuronal viability. Data points represent average % viability relative to vehicle ± SEM for $n = 3$ biological replicates. Source data are provided as a Source Data file.

4 days of treatment, with recovery starting at day 8 up to near basal levels by day 16–20 (Fig. 8g, l, q).

**Exposure-activity relationship for mTORi and tau regulation.** Prolonged effect on mTORC1 activity could be a result of compound stability within the cell and/or extended target engagement. Although the assays employed cannot resolve this, we asked how much compound penetrates neuronal cells or remains in the extracellular media due to permeability issues, non-specific binding, or cell membrane efflux transporters. As a case study, we

treated tau-WT neurons with 1 μM or 10 μM AZD2014 and measured compound concentration in cells and in media (Supplementary Fig. 9c). Equilibrium dialysis was also performed on 10 μM-treated samples to determine what percentage of AZD2014 was present as free drug. Since free drug is a function of drug and matrix properties, it is typically not dose-dependent. Of the total concentration added to neuronal cultures, 20% of the 10 μM dose and 40% of the 1 μM dose were detected intracellularly (Supplementary Fig. 9c). By equilibrium dialysis it was determined that for the 10 μM dose ~4.5% was free compound. Thus, a 10 μM dose of AZD2014 for 24 h produced a total cellular concentration

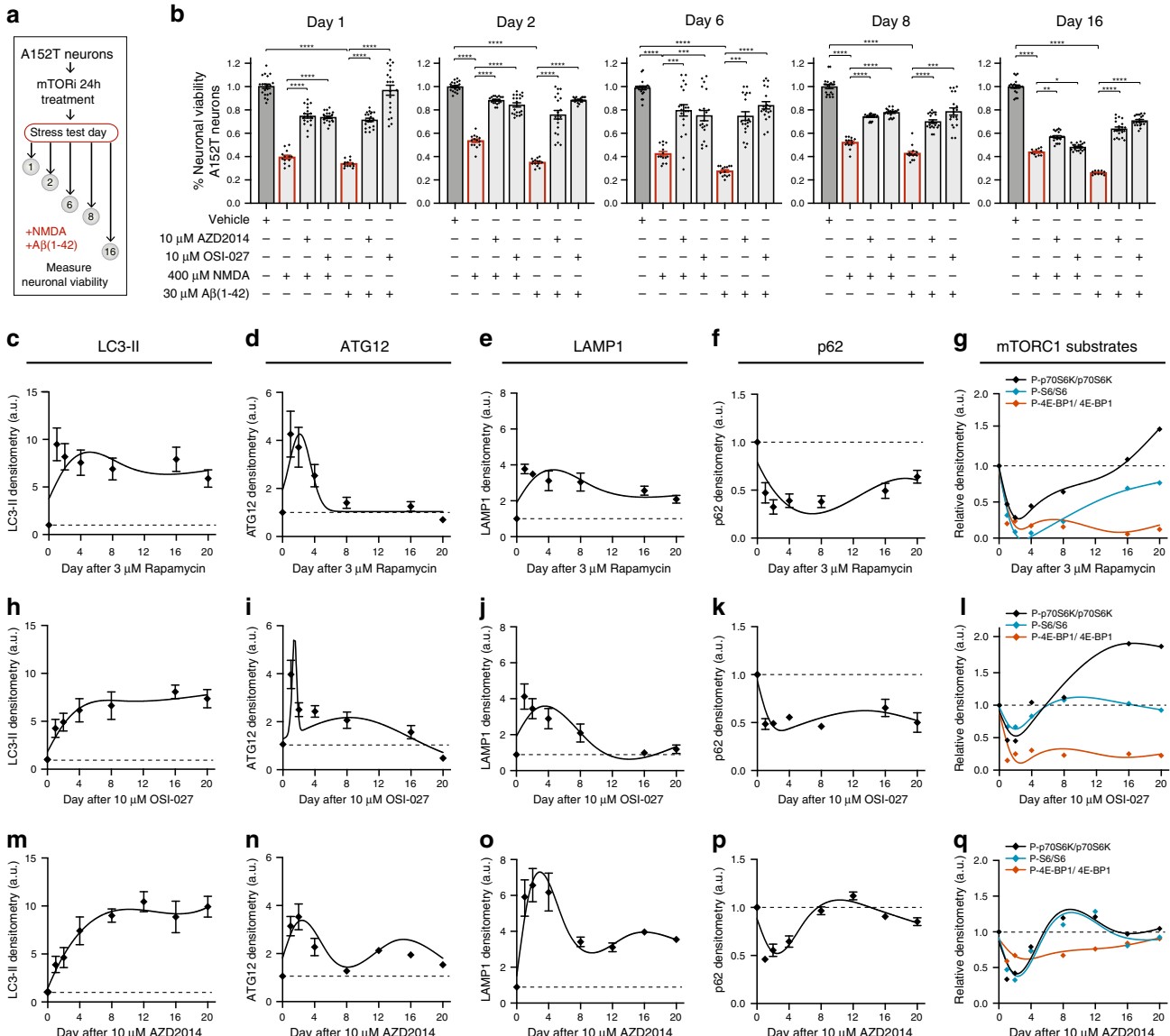

**Fig. 8 Time-course analysis of mTORi effect on neuronal vulnerability to stress, autophagy and mTORC1 activity. a** Assay to measure mTORi time-course effect on neuronal vulnerability to stress. A152T neurons (FTD19-L5-RC6, 6 weeks differentiated) were treated with 10 µM AZD2014 or OSI-027 for 24 h (day 0–1), followed by media change. Then, on each day indicated, neurons were stressed with either 400 µM NMDA or 30 µM Aβ(1-42) for 18 h, and viability was measured. **b** Graph bars represent mean % viability, relative to vehicle ± SEM and diamond dots represent individual data points for $n = 3$ (and 4 technical replicates per experiment). Statistical significance was calculated with a two-tailed unpaired $t$-test (*$P \leq 0.05$, **$P \leq 0.01$, ***$P \leq 0.001$, ****$P \leq 0.0001$). **c–q** Neurons were treated with 3 µM rapamycin (**c–g**), 10 µM OSI027 (**h–l**) or 10 µM AZD2014 (**m–q**) and analyzed by western blot over a period of 20 days (Supplementary Fig. 9a, b). **c–f, h–k, m–p** Time-course densitometry of LC3-II, ATG12, LAMP1 and p62 autophagy protein levels. Data points represent mean densitometry, relative to vehicle-treated samples (dotted lines) ± SEM for $n = 3$ biological replicates. **g, l, q** Time-course measure of mTORC1 substrates' phosphorylation (p70S6K, S6, 4E-BP1). Data points represent mean densitometry, relative to vehicle-treated samples (dotted lines) for $n = 2$ biological replicates. Source data are provided as a Source Data file.

of 1,975 ± 583 nM with an estimated free drug concentration of ~89 nM, which is ~30-fold above the demonstrated 2.8 nM IC$_{50}$ for in vitro mTOR inhibition[66]. By extrapolation, for the 1 µM dose this would project a free drug concentration of ~17 nM, which is sixfold above the in vitro mTOR IC$_{50}$ for AZD2014. At the concentrations tested here, it appears that a much lower free drug concentration led to an effect on neuronal tau. For example, for the neuronal AZD2014 EC$_{50}$ 100 nM, the effective free drug concentration would be estimated at ~1–2 nM, around the effective in vitro IC$_{50}$ of the drug (2.8 nM)[67].

In order to interpret these results in an in vivo context, we needed to understand AZD2014 permeability and distribution in

the central nervous system (CNS), and estimate the potential free brain concentrations achievable in the clinic. To determine the extent of AZD2014 CNS penetration, rats were continuously infused with AZD2014 for 4 h, followed by determination of brain and plasma concentrations. The free brain concentrations achieved were ~9% of the free plasma concentrations, indicating that whilst AZD2014 distributed into the brain, its CNS penetration was sub-optimal. In vitro work by others has shown that AZD2014 is a substrate of two efflux transporters, P-glycoprotein (P-gp) and Breast Cancer Resistance Protein (BCRP), both expressed at the blood-brain barrier and likely to limit drug penetration into the brain[68,69]. Utilizing available

clinical data for AZD2014[66] and applying the unbound partition coefficient determined in rat brain, we estimated that, at the highest clinical dose tolerated with intermittent dosing regimen, the maximum achievable free concentration in the human brain would be 32 nM. This concentration is within the range of free concentrations (~1–89 nM, Supplementary Fig. 9c) shown to have significant effect in our ex vivo human tauopathy model (Figs. 2g, 6c, and 7d, f).

## Discussion

In this study we established that tau from FTD patient-derived neurons is amenable to degradation via autophagy, leading to rescue of neuronal stress vulnerability. We characterized the mechanism-of-action of three mTORi compounds OSI-027, AZD2014, AZD8055 (Supplementary Table 1), that upregulated autophagy in healthy and tauopathy neurons. We demonstrated that tau reduction is dependent on autophagy (Fig. 4), through sequestration into phagolysosomes (Fig. 5, Supplementary Fig. 6), and as a consequence of mTORC1 inhibition (Fig. 6). In this neuronal model, mTORi reduced tau without clear discrimination between total tau and P-tau (Fig. 2a–h, Supplementary Fig. 3a–d), but with preference for insoluble tau species (Fig. 3b–d) predicted to represent misfolded and aggregation-prone tau. Our most notable and novel finding was the discovery that a single 24h-dose of AZD2014 or OSI-027 caused reduction of tau for 12–16 days post treatment without loss of cell viability or integrity (Fig. 7c–h), and across independent neuronal models (Supplementary Fig. 8). Moreover, we found an increase in neuronal resistance to stress (Fig. 8b), which indicates that tau toxicity was suppressed. Prolonged tau reduction coincided with mTORC1 inhibition and increased autophagy activity, further supporting our proposed model (Supplementary Fig. 1). Therefore, even though the ALP might be involved in disease, its function can be pharmacologically enhanced to offer protection against tau pathogenicity.

Current research on therapeutics for neurodegeneration has put forward several candidates[5,44,70–72], including autophagy modulators[5,12,14,22,25,26,28–31], but there is still a lack of evidence for efficacy in human brain at patient tolerated doses. To our knowledge, our work is the first screen and demonstration of pharmacological upregulation of autophagy that rescues phenotypes in tauopathy patient-derived neurons. Given the failure in clinical trials for AD and related dementias, pre-clinical research solely in humanized animal models is not sufficient[73]. Introducing patient-derived neuronal models that recapitulate early aspects of tau pathology ex vivo in a drug discovery pipeline is very beneficial. These neurons allow access to physiologically relevant cell types, protein complexes, and disease-associated genomic background, without the need for overexpression of heterologous genes[39,40,74].

OSI-027, AZD2014, and AZD8055 are orally available, potent, and specific mTOR inhibitors, affecting both mTORC1 and mTORC2 complexes[32], but in human neurons their effect had not been thoroughly investigated. Here, these mTORi had a stronger effect on tau clearance than rapamycin, without affecting viability. However, in tumor cells and organ transplant studies and clinical trials, all these compounds have shown a plethora of side-effects[10,32–36]. Although this poses an obstacle for treatment of older patients with neurodegenerative diseases[67,75], adverse effects are usually dose- and frequency-dependent, and reversible upon treatment interruption[32]. Moreover, there is now increased interest on mTORC1/2 inhibitors for positively impacting hallmarks of aging in animal models[42,67]. Our study also shows that OSI-027 and AZD2014 had a prolonged effect on tau levels and toxicity (12–16 days) after a single dose (Fig. 7), without reducing cell viability. Therefore, we propose that adverse effects might be counter-balanced by an intermittent dosing regimen, on account of prolonged drug effect. Also, increased tolerability might be achievable under optimal efficacy doses that still retained a prolonged effect (Fig. 7c, d vs. e, f).

Despite our lead compounds being described as dual mTORC1/2 inhibitors from studies in vitro and in cancer cells, we found that tau clearance in neurons was mainly achieved through mTORC1 inhibition (Fig. 6b–i), with less than a 10% effect on P-AKT$^{S473}$ levels (Supplementary Fig. 7b, c). Also, in neurons rapamycin reduced the phosphorylation of 4E-BP1$^{T37/46}$ (Fig. 8g), a substrate of mTORC1 previously shown to be rapamycin-resistant in cancer cells[32]. It was not surprising to find differences in molecular targets between proliferative cancer cells and post-mitotic neurons[32].

Unfortunately, clinical efficacy of mTORC1/2 inhibitors for brain malignancies has also been limited by poor drug brain permeability[69,76]. This may be caused by efflux transporters at the blood-brain barrier, such as multi-drug resistance proteins that restrict small-molecules penetration[68,69]. Here, we show that, even though AZD2014 is a substrate of both P-gp and BCRP, it penetrates the rat brain and is estimated to achieve free brain concentrations at tolerated doses in humans that are equivalent to the range of free concentrations measured in ex vivo human neurons that rescued tauopathy phenotypes.

Altogether, our study supports a therapeutic potential for mTORi that promote autophagy-mediated tau clearance, that could also be beneficial for a number of other proteinopathy diseases.

## Methods

**Human NPC lines.** Approval for work with human subjects and derived iPSCs was obtained under the Massachusetts General Hospital/Partners Healthcare-approved IRB Protocol (#2010P001611/MGH). Cell lines employed in this study were derived from a male individual in his 60 s carrying the tau risk variant A152T (c.1407 G > A NCBI RefSeq NM_001123066, rs143624519), diagnosed with PSP, and categorized as cell line FTD19-L5-RC6;[21] from a female individual in her 50 s carrying the autosomal dominant mutation P301L (c.C1907T NCBI NM_001123066, rs63751273), diagnosed with FTD and categorized as cell line MGH2046-RC1;[37,44] from a male individual in his 60 s, unaffected control tau-WT, categorized as cell line Control-1 or 8330-8-RC1;[21] and from a female individual in her 40 s unaffected control tau-WT, categorized as cell line Control-2 or MGH2069-RC1[37,44]. Briefly, fibroblasts were reprogrammed into iPSCs by non-integrating methods, which were subsequently converted into cortical-enriched neural progenitor cells (NPCs) and differentiated into neuronal cells as previously described[21,37]. Generation and characterization of the polyclonal MAPT knock-down (MAPT-Kd1-3) NPC lines via CRISPR/Cas9 engineering of the parental line FTD19-L5-RC6 was previously described[21].

**Cell culture and differentiation.** NPCs were cultured in six-well (Fisher Scientific Corning) or black 96-well clear bottom (Fisher Scientific Corning) plates coated with poly-ornithine (20 μg/mL in water, Sigma) and laminin (5 μg/mL in PBS, Sigma), referred to as POL-coated plates, in DMEM/F12-B27 cell media [70% DMEM (Gibco), 30% Ham's-F12 (Fisher Scientific Corning), 2% B27 (Gibco), 1% penicillin-streptomycin (Gibco)]. Media was supplemented with EGF (20 ng/mL, Sigma), FGF (20 ng/mL, Stemgent) and heparin (5 μg/mL, Sigma) to promote NPC proliferation. For NPC differentiation, growth factors were omitted from the cell media and cells were cultured for a period of several weeks (experiment-dependent) with change of half-volume culture media twice per week[21].

**Compounds and antibodies.** Listed with respective commercial information in Supplementary Table 2.

**Microscopy LysoTracker and CYTO-ID small-molecule screen.** For the primary screen with Control-1 (8330-8-RC1) NPCs, cells were plated in 96-well plates at a starting density of 47,000 cells/cm$^2$ for 24 h in DMEM/F12-B27 cell media with growth factors. Compounds from a chemogenomic library of experimental and FDA-approved drugs (+500 nM CQ) were added directly into the media and incubated at 37 °C for 24 h. Low dose CQ was only used for the LysoTracker/CYTO-ID microscopy screens. DMSO (Sigma) final concentration was kept at < 0.05% (v/v). NPCs were co-stained with LysoTracker Red DND-99 (Life Technologies L-7528) and CYTO-ID Green (Enzo Life Sciences ENZ-51031), according to manufacturer's protocols. Briefly, LysoTracker was added at 500 nM and

incubated for 45 min at 37 °C. Cells were then washed with Assay Buffer and incubated with Microscopy Dual Detection Reagent, which includes CYTO-ID green and Hoechst 33342 dye, for 30 min at 37 °C. Cells were washed again with Assay Buffer and fixed in 4% (v/v) formaldehyde-PBS (Tousimis) for 30 min at room temperature. Automated image acquisition was done with the InCell Analyzer 6000 Cell Imaging System (GE Healthcare Life Sciences) with 15 fields acquired per well ($n = 5$ biological replicates). Image analysis was done with CellProfiler version 2.2.0 (free open-source software), consisting of quantification of number of fluorescent puncta per nuclei, corrected for intensity. Further calculations were done using Microsoft Excel version 16.36 and graphs were plotted in GraphPad Prism 8 version 8.4.2. For the secondary screen with Control-1 (8330-8-RC1) neurons, NPCs were plated in 96-well plates at a starting density of 110,000 cells/cm$^2$ in DMEM/F12-B27 media and differentiated for 5 weeks, with half-volume of culture media changed twice a week. Compounds (+500 nM CQ) were added directly onto the media and incubated for 24 h at 37 °C. DMSO final concentration was kept at < 0.05% (v/v). Neuronal cells were stained with Lyso-Tracker and CYTO-ID separately, fixed in 4% (v/v) formaldehyde-PBS (Tousimis) for 30 min, and stained with nuclear dye Hoechst 33342 (1:2500) and neuronal marker MAP2 (1:1000) according to the IF protocol and Supplementary Table 2. Automated image acquisition was done with the InCell Analyzer 6000 Cell Imaging System (GE Healthcare Life Sciences), for $n = 3$. Micrographs were assembled using Adobe Photoshop 2020 version 21.1.2.

**Immunofluorescence of neuronal cells.** NPCs were plated at a starting density of 110,000 cells/cm$^2$ in black POL-coated 96-well clear-bottom plates (Corning), in DMEM/F12-B27 media and differentiated for six weeks, followed by compound treatment. Neurons were fixed with 4% (v/v) formaldehyde-PBS (Tousimis) for 30 min, washed in PBS (Corning), incubated in blocking/permeabilization buffer [10 mg/mL BSA (Sigma), 0.05% (v/v) Tween-20 (Bio-Rad), 2% (v/v) goat serum (Life Technologies), 0.1% Triton X-100 (Bio-Rad), in PBS] for 2 h, and incubated with primary antibodies overnight (Supplementary Table 2: P-Tau PHF1 1:1000, Tau K9JA 1:1000, MAP2 1:1000, Hoechst 33342 1:2500). Cells were washed with PBS and incubated with the corresponding AlexaFluor-conjugated secondary antibodies at a 1:500 dilution (Life Technologies, Supplementary Table 2,). Image acquisition was done with the InCell Analyzer 6000 Cell Imaging System (GE Healthcare Life Sciences) and micrographs were assembled using Adobe Photoshop 2020 version 21.1.2.

**Neuronal viability.** Assay for neuronal cells cultured in black 96-well plates with a clear bottom (Fisher Scientific Corning), as previously described[21]. Viability was measured with the Alamar Blue Cell viability reagent (Life Technologies) at 1:10 dilution, after 4 h incubation at 37 °C, according to manufacturer's instructions. Readings were done in the EnVision Multilabel Plate Reader (Perkin Elmer). Calculations were done using Microsoft Excel version 16.36 and graphs were plotted in GraphPad Prism 8 version 8.4.2.

**Compound treatment for protein and mRNA analysis.** NPCs were plated at an average density of 75,000 cells/cm$^2$ of six-well POL plates in DMEM/F12-B27 media and differentiated for 5–10 weeks, depending on the experiment. Compound treatment was performed by removing half-volume of neuronal-conditioned media from each well and adding half-volume of new media pre-mixed with the compound at 2X final concentration, followed by incubation at 37 °C for the designated period of time. When performing drug pre-treatment, the first compound was added as described above, for a period of time, and then mTORi was added directly into the media without further media exchange.

**Protein purification and western blot.** Neurons were washed and collected in PBS (Corning), lysed in RIPA buffer (Boston Bio-Products) supplemented with 2% SDS (Sigma), protease inhibitors cocktail (Roche Complete Mini tablets), and phosphatase inhibitors cocktail (Sigma), followed by 5 min water sonication (Bransonic Ultrasonic Baths, Thomas Scientific) and 20,000 × $g$ centrifugation for 20 min. Supernatants were transferred to new tubes, protein concentration was determined with the Pierce BCA Protein Assay Kit (ThermoFisher Scientific), and 10 μg of total protein in SDS loading buffer (NEB) were analyzed by SDS-PAGE. Alternatively, cell lysis was performed directly in 5x the cell pellet volume with SDS sample loading buffer, and boiled for 15 min prior to SDS-PAGE (Figs. 6j, 7b–f, and Supplementary Figs. 5b-d, 7a, 8, 9a, b). Electrophoresis was performed with the Novex NuPAGE SDS-PAGE Gel System (Invitrogen). Proteins were transferred from the gel onto PVDF membranes (EMD Millipore) using standard procedures. Membranes were blocked in 5% (w/v) BSA (Sigma) in Tris-buffered saline with Tween-20 (TBST, Boston Bio-Products), incubated overnight with primary antibody (Supplementary Table 2: antibody information and dilution) at 4 °C, followed by corresponding HRP-linked secondary antibody at 1:4000 dilution (Cell Signaling Technology). Blots were developed with SuperSignal West Pico Chemiluminescent Substrate (ThermoFisher) according to manufacturer's instructions, exposed to autoradiographic films (LabScientific by ThermoFischer), and scanned on an Epson Perfection V800 Photo Scanner. Protein bands' densitometry (pixel mean intensity in arbitrary units, a.u.) was measured with Adobe Photoshop 2020 version 21.1.2 Histogram function and normalized to the respective internal

control (β-Actin or GAPDH) band. Graphs and statistical analysis were done in GraphPad Prism 8 version 8.4.2.

**Tau ELISA.** Neuronal cells were collected and lysed in ELISA-compatible buffer (Invitrogen) supplemented with 1 mM PMSF, protease (Roche) and phosphatase (Sigma) inhibitors, and incubated for 30 min on ice. Lysates were centrifuged at 18,000 × $g$ at 4 °C for 10 min. Cleared lysates' total protein concentration was determined with the Pierce BCA Protein Assay Kit (ThermoFisher Scientific). ELISA was performed according to manufacturer instructions, with the Human Total Tau ELISA Kit (Invitrogen) and the P-Tau[pS396] Human ELISA Kit (Invitrogen). Spectrophotometric measures were done in a Spectra-max Plus 384 (Molecular Devices) plate reader. Calculations were done using Microsoft Excel version 16.36, graphs and statistical analysis were done in GraphPad Prism 8 version 8.4.2.

**Protein solubility analysis.** Protein fractionation based on detergent solubility was performed as previously described (Fig. 3a)[21,45,46]. Briefly, higher solubility proteins were purified in 1% (v/v) Triton X-100 buffer (T fractions), whereas lower solubility (insoluble) pelleted proteins were resuspended in 5% (v/v) SDS buffer (S fraction). SDS-PAGE western blot was performed by loading 15 μg of T-fraction and equal volume of the S-fraction onto pre-cast SDS-PAGE (Novex, Invitrogen). Densitometry values (pixel mean intensity in arbitrary units, a.u.) were measured with the Histogram function of Adobe Photoshop 2020 version 21.1.2 and normalized relative to the respective GAPDH intensity in T-fraction. Calculations were done in Microsoft Excel version 16.36; graphs were plotted in GraphPad Prism 8 version 8.4.2 and statistical analysis done in Microsoft Excel version 16.36.

**RNA extraction and quantitative RT-PCR.** Cells were collected in PBS, pelleted at 3000 × $g$ for 5 min, and lysed with Trizol reagent (Ambion Life Technologies 15596018). Total RNA was extracted using the Direct-zol RNA MiniPrep (ZYMO Research R2052), according to the manufacturer's instructions. RNA eluted from the Zymo-Spin IICR Column was quantified with a Nanodrop spectrophotometer (Thermo Scientific). RNA was reverse transcribed using the High Capacity cDNA Reverse Transcription Kit (Applied Biosystems, ThermoFisher Scientific 4368814) according to the manufacturer's protocol, using a standard thermal cycler (Bio-Rad). Quantitative real-time PCR (qRT-PCR) was performed using the TaqMan Assay and TaqMan Gene Expression Master Mix (ThermoFisher Scientific), in 384-well plate format (Corning Axygen), as per manufacturer's specifications, using 1 μL of cDNA sample at 1:1 and 1:10 dilutions. Real-time PCR was run in the LightCycler 480 II (Roche) instrument. The TaqMan gene expression assay probes used were: *MAPT* (8 RefSeq NM) Hs00902193_m1, *MAPT* (2 RefSeq NM) Hs00902978_m1, *CTSD* (1 RefSeq NM) Hs00157205_m1, *LAMP1* (1 RefSeq NM) Hs00931462_m1, *SQSTM1* (3 RefSeq NM) Hs01061917_g1, *ACTB* (1 RefSeq NM) Hs03023943_g1, *GAPDH* (2 RefSeq NM) Hs99999905_m1 (Thermo Fisher Scientific). Relative quantification of gene expression was done using the Comparative C$_T$ Method ($\Delta\Delta$C$_T$ Method), based on C$_T$ (cycle threshold) values for each amplification curve. Expression of actin (*ACTB*) and *GAPDH* were used as internal controls. For *MAPT* expression, fold-change values obtained with each PCR probed were averaged. Calculations and statistics were done in Microsoft Excel version 16.36; graphs were plotted in GraphPad Prism 8 version 8.4.2.

*MAPT* mRNA expression analysis for the *MAPT*-Kd lines (Supplementary Fig. 2i), has been previously described[21]. The primers used were total *MAPT* Fw 5-CAAGCTCGCATGGTCAGTAA-3, Rev 5-CAGAGCTGGGTGGTGTCTTT-3; and for *GAPDH* Fw 5-CCATGGCACCGTCAAGGCTGA-3, Rev 5′-GCCAGTAG AGGCAGGGATGAT-3′.

**Stress vulnerability assays.** Performed as previously described[21] and summarized in Supplementary Fig. 4f. NPCs were plated (110,000 cells/cm$^2$) and differentiated in 96-well plate format, for 8 weeks. mTORi or DMSO (Sigma) were added directly onto the media and incubated for 8 h at 37 °C. Then, each well was treated with 30 μM Aβ(1-42), 400 μM NMDA (Supplementary Table 2) or vehicle alone, for an additional 16 h. At 24 h, viability was measured with the Alamar Blue Cell Viability Reagent (Life Technologies) and with the EnVision Multilabel Plate Reader (Perkin Elmer). Calculations and statistics were done in Microsoft Excel version 16.36; graphs were plotted in GraphPad Prism 8 version 8.4.2.

**Sub-cellular fractionation of lysosomes.** Lysosomes isolation was carried out with the Lysosome Enrichment Kit for Tissue and Cultured Cells (Thermo Scientific), according to the manufacturer's protocol[77]. Briefly, ~ 25 million differentiated neurons (pellet sizes 80–100 mg) were used for each 8h-compound treatment and purification, with $n = 2$ biological replicates for each cell line. Lysis buffers were supplemented with Halt protease and phosphatase inhibitors cocktail (Thermo Scientific). After lysis in Lysosome Enrichment Reagent A + B, samples were centrifuged at 500 × $g$ for 10 min and an aliquot of the supernatant was kept as a sample of the cytosolic fraction (total protein minus nuclei). The remaining lysate was mixed with OptiPrep Cell Separation Media and overlaid onto an OptiPrep density gradient. Samples were ultra-centrifuged at 145,000 × $g$ for 2 h at 4 °C, and the lysosome fractions (top gradient band) were transferred to eppendorf tubes and kept on ice. Lysosomes were washed and pelleted at 18,000 ×$g$

for 30 min at 4 °C, and then lysed in 2% (w/v) CHAPS detergent in Tris-buffered saline (Thermo Scientific), supplemented with Halt protease and hosphatase inhibitors cocktail (Thermo Scientific), and centrifuged at $18,000 \times g$ at 4 °C for 5 min. The clarified supernatants containing lysosomal proteins were analyzed by SDS-PAGE western blot (Supplementary Table 2: antibodies and dilutions).

**Exosome preparation.** Neuronal culture media was cleared by centrifugation at $3000 \times g$ for 15 min to remove cells and cell debris, and then supplemented with Halt protease and phosphatase inhibitors cocktail (Thermo Scientific). Each cleared supernatant was incubated with ExoQuick-TC Exosome Precipitation Solution (EXOQ EXOTC10A-1) according to manufacturer's instructions[78]. Briefly, 2 mL of ExoQuick-TC solution were added per 10 mL of cell media, mixed well by inverting, and incubated at 4 °C overnight. The resultant suspensions were centrifugated at $1500 \times g$ for 30 min at 4 °C. Each exosomal pellet was resuspended in SDS sample loading buffer (NEB) and analyzed by SDS-PAGE western blot (Supplementary Table 2 antibodies and dilutions used).

**Time-course recovery assays.** Neurons were differentiated for 6 weeks and on day 0, vehicle alone (DMSO) or mTORi was added to the neuronal cultures at 100 nM or 10 μM (3 μM for rapamycin) and incubated for 24 h at 37 °C. DMSO was kept at ≤0.1% (v/v). On day 1, cells were washed and new cell media without compound was added. At each time point between day 1 and day 20, one well of cells corresponding to each mTORi treatment was sacrificed for viability measurement followed by protein analysis. For stress vulnerability assays, on each stress test day, each well corresponding to mTORi pre-treated neurons, was sacrificed and either 30 μM Aβ(1-42) or 400 μM NMDA were added to the cell media for 16 h. Cell viability was measured with the Alamar blue assay as described above and calculated as a relative percentage of vehicle-alone treated neurons (100%).

**Micro-dialysis and mass spectrometry.** Control-1 (8330-8-RC1) NPCs were plated at ~75,000 cells/cm² in 6-well plates, differentiated for 6 weeks and treated with 1 μM or 10 μM AZD2014 for 24 h. Cell media was collected into ice-cold eppendorf tubes and centrifuged at $3000 \times g$ for 10 min at 4 °C. Neurons were washed twice with ice-cold PBS (Corning), transferred to eppendorf tubes and pelleted at $3000 \times g$ for 5 min. Each pellet corresponded to three wells of cells, and each treatment was done in triplicate (n = 3). Cells were lysed in RIPA buffer (Boston Bio-Products) supplemented with HALT protease and phosphatase inhibitor cocktail (Thermo Fisher) and 1:5000 Benzonase (Sigma). Lysates were centrifuged at $20,000 \times g$ for 15 min and the supernatants were used for protein quantification (BCA assay) and analysis. UHPLC-MS-MS analysis was performed using an Agilent 1290 UHPLC system coupled to an Agilent 6490 triple quadrupole mass spectrometer. The chromatographic separation used an Acquity HSS T3 column, 2.1 ×50 mm, 1.8 μm i.d. (Waters) and a binary gradient with eluent A = 0.1% formic acid in water and eluent B = 0.1% formic acid in acetonitrile (all solvents were of LC/MS grade). The gradient program had an initial condition of 5% B, which was ramped to 75% B over 2 min, then held at 75% B for 0.5 min, then held at 5% B for 0.5 min for re-equilibration. The flow rate was 0.8 mL/min, the column was at room temperature, and the injection volume was 2 μL. MS acquisition was carried out with an electrospray ionization (AJS ESI) ion source operated in positive mode. Source parameters were as follows: gas temperature 200 °C, gas flow 19 L/min, nebulizer pressure 45 psi, sheath gas temperature 225 °C, sheath gas flow 11 L/min, capillary 2500 V, nozzle voltage 0 V. MRM transitions monitored were 463.25 precursor to 405.3 product, collision energy 40 V (AZD2014 quantifier); 463.25 precursor to 347.3 product, collision energy 40 V (AZD2014 qualifier); and 327.14 precursor to 270.1 product, collision energy 18 V (clozapine internal standard quantifier). Absolute quantification of AZD2014 was performed by external calibration. Internal standard of clozapine (Sigma) was prepared at 1 μM in DMSO. AZD2014 standards (Selleck) were prepared ranging from 10 nM to 10 μM in 1:1 water:acetonitrile and then 10 μL of each standard was diluted with 90:5 RIPA buffer:internal standard. Cell lysate samples were prepared for analysis by mixing 45 μL sample with 5 μL RIPA buffer and 2.5 μL internal standard. A portion of each lysate/supernatant from the 10 μM incubation samples was subjected to overnight dialysis using a Rapid Equilibrium Dialysis (RED) Device (Thermo Fisher). 50 μL of lysate and 50 μL of PBS (Quality Biological) were placed in the sample compartment and 300 μL of PBS in the buffer compartment, then placed on an orbital shaker for 18 h at room temperature. Post dialysis, 50 μL aliquots were taken from each compartment, then 50 μL RIPA and 50 μL acetonitrile and 7.5 μL internal standard were added. All quantitation was performed using Agilent MassHunter software (version B.07.01) to determine the absolute concentration of AZD2014 in each sample. For each post dialysis pair of samples, the ratio of AZD2014 concentration in the buffer compartment to the sample compartment was used to determine the % free compound, since only free compound migrated to the buffer compartment during dialysis, while the sample compartment contained both free and bound compound.

**AZD2014 CNS penetration study in rat.** n = 3 Han Wistar male rats were purchased from Vital River (Beijing, China) by Pharmaron (Beijing, China) and studies were performed under Institutional Animal Care and Use Committee (IACUC)-approved protocol #PK-R-06012018. Animals were 8–10 weeks old and weighted between 280 and 300 g. AZD2014 was administered as an intravenous infusion for 4 h at a dose of 2 μmol/kg/h in a total dose volume of 4 mL/kg in TEG: DMA:Water (1:1:1). At the end of infusion, 0.4 mL of blood were collected by heart puncture. Samples were transferred into plastic micro centrifuge tubes containing EDTA-K2 as anticoagulant and centrifuged at $4000 \times g$ for 5 min at 4 °C to obtain plasma. Rats were fully exsanguinated prior to brain collection. Brain samples were placed in tared tubes, weighed, and then purified water was added at a 1:3 ratio of brain weight (g): water volume (mL), followed by homogenization. Concentrations of AZD2014 in the plasma and brain samples were determined using liquid chromatography-tandem mass spectrometry (LC-MS/MS). For brain samples, the final brain concentration was the detected value multiplied by the dilution factor and from which 0.8% of total plasma concentration was subtracted to account for residual blood. The CNS penetration was measured by calculating the coefficient of partition of the unbound (free) brain concentrations to the unbound (free) plasma concentrations ($Kp_{u,u}$), i.e., the ratio of free brain concentrations to free plasma concentrations. Free brain concentrations were derived from measured total brain concentrations, multiplied by the free fraction in brain determined in a brain slice distribution, as described by Friden et al.[79]. Brain samples were tested in singlicate at the highest dilution factor (9) and in duplicate at other dilution level (onefold and threefold) within the same bioanalytical run. The values used were the results from the less diluted samples, giving the most robust results and with all samples within a dilution group within the quantification limits. For samples run in duplicate, the results were considered the most robust if all repeats were within 20% of the first result and the first result was used in the calculations. Free plasma concentrations were derived from total plasma measured concentrations multiplied by the free fraction in plasma, determined by rapid equilibrium dialysis. Plasma samples were tested in singlicate at three different dilutions (1, 3, and 9-fold dilution) within the same bioanalytical run. The values used were the results from the less diluted samples, as long as all samples in that dilution group were within the quantification limits.

**Statistical information.** Data are presented as mean values ± SD (standard deviation) or ±SEM (standard error of the mean), calculated using Microsoft Excel 2016/version 16.36 and GraphPad Prism 8 version 8.4.2. P value < 0.05 was considered the threshold for statistical significance. P value significance intervals (*) are provided within each figure legend, together with the statistical test performed for each experiment: unpaired two-tailed Student's t-test calculated in Microsoft Excel version 16.36 or two-way ANOVA followed by post hoc test for multiple comparisons calculated in GraphPad Prism 8 version 8.4.2. N values are also indicated within figure legends and refer to biological replicates (NPC or neuronal cultures independent setup and analysis, at different times), whereas technical replicates refer to repeated analysis of the same samples. Derived statistics correspond to analysis of averaged values across biological replicates, and not pooled technically and biological replicates.

**Reporting summary.** Further information on experimental design is available in the Nature Research Reporting Summary linked to this article.

## Data availability

A reporting summary for this article is available as Supplementary Information file. The main data supporting the findings of this study are available within the article and its Supplementary Figures. The source data underlying Figs. 1–4, Figs. 6–8, Supplementary Figs. 2–5 and Supplementary Figs. 7–9 are provided as a Source Data file. Specific data P values are also included within the Source Data file. Additional details on datasets and protocols that support the findings of this study will be made available by the corresponding author upon reasonable request. Source data are provided with this paper.

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

## Acknowledgements

We wish to thank Dr. Sabina Cosulich (AZ Cambridge Biomedical Campus, Cambridge UK) and Dr. Raymond Chen (AZ R&D Principal Scientist) for critical discussions and feedback; and Dr. Peter Davies (Albert Einstein College of Medicine, NY) for the Tau PHF-1 antibody. We wish to thank members of the Tau Consortium (Rainwater Charitable Foundation) for helpful feedback on tauopathy iPSC-derived cell models and autophagy pathway biology. M.C.S. was a fellow of the AstraZeneca postdoctoral program and received a postdoctoral fellowship award from the Association for Frontotemporal Degeneration (AFTD). S.J.H. laboratory has received relevant funding from the Tau Consortium, National Institutes of Health (R21 NS085487), Bluefield Project to Cure FTD, and the MGH Research Scholars Program.

## Author contributions

M.C.S., N.J.B., and S.J.H. contributed with conception and design of the work. M.C.S. executed the experimental work, data acquisition and analysis. G.N. contributed with cell culture work and data acquisition. S.T. performed the equilibrium dialysis, mass spectrometry experiment and data analysis. I.K.G and T.J. contributed with the animal work and data analysis. D.L. and B.C.D. contributed with patient recruitment, acquisition of the tau-P301L patient fibroblasts, and manuscript revision. M.C.S., D.G.B., N.J.B., and S.J.H. contributed with project supervision and data interpretation. M.C.S. wrote the manuscript. D.G.B., N.J.B., and S.J.H. substantially revised the manuscript and approved the submitted version.

## Competing interests

At the time of this study, S.T., I.K.G., T.J., D.G.B. and N.J.B. were fulltime employees by and shareholders in AstraZeneca. S.J.H. is a member of the SAB and equity holder in Rodin Therapeutics, Psy Therapeutics, Frequency Therapeutics, and Souvien Therapeutics, none of whom were involved in this study. S.J.H. has received consulting or speaking fees from Sunovion, Biogen, AstraZeneca, Amgen, and Merck. B.C.D. is a consultant with Arkuda, Axovant, Biogen, Lilly, Merck, Novartis and Wave Life Sciences; and has editorial duties with honoraria with Elsevier. B.C.D. also received royalties from Oxford University Press and Cambridge University Press. None of these entities were involved in this study.
