## [Peer Review File · Nature Communications]

Reviewers' comments:

Reviewer #1 (Remarks to the Author):

In this paper, the authors examine tau reduction via inhibition of mTOR signalling as a potential tau therapeutic in iPSC-neurons. The authors show that a single dose can lower tau protein levels for up to 12 days following treatment, and this reduces the susceptibility of neurons to stressors such as ABeta. Tau reduction is a promising avenue as a potential therapeutic, with increasing evidence that tau reduction can be protective against amyloid toxicity, and tau ASO are already in clinical trials. Thus, the data will be of broad interest to the field. The data are well presented and the manuscript is well-written and easy to follow. Some points minor points to be addressed:

- The authors suggest an increase in components of the ALP machinery specifically in mutant vs. control neurons (Figure S1). However, it is hard to appreciate this as each line has a separate, spliced panel – and so difficult to assess whether they are different exposures etc. It would be useful to include samples from each line run on a single gel – e.g. n=3 from each line at 12 weeks on the same gel to allow the reader to appreciate the change and any variability.
- The authors routinely include a side panel of recombinant tau ladder beside the tau western blots, but this is redundant as 1. It is a spliced lane and not part of the main gel 2. The samples are not dephosphorylated and so the molecular weight will be a reflection of many factors including phosphorylation state 3. As a recombinant tau ladder, one assumes the ladder itself is dephosphorylated and therefore wouldn't be detected by the pTau 396 antibody? I would suggest removing the ladder throughout as it doesn't add to the figures or help interpretation.
- The banding patterns observed with the pTau 396 antibody are highly variable between lines, time points and experiments. For the quantification of these blots, the authors use some higher molecular weight bands - can they provide evidence these bands are tau species and not non-specific – especially as they do not appear to be detected by the Tau5 antibody?
- The prolonged decrease of tau levels following mTORi is a very interesting finding by the authors. However, the western blots presented in Figure S8 show some variability between compounds and experiments – e.g. pTau396 in the 10M AZD2014 appear to decrease at the later timepoints only, total tau in the OSI-027 treated cells fluctuates a little. I think it would help the reader to have an example western blot for each compound included the main figure, and all 3 replicates as supplementary data.
- Although the treatments do not appear to reduce neuronal viability, the cells used will still be in a neurodevelopment phase and I wonder if tau reduction could be having more subtle effects on neurodevelopment – for example reduced neurite outgrowth. It would be beyond the scope of the current manuscript to look at this, however it would be good to include this in the discussion and I think it would also be good to include western blot data with tau, actin, and the additional loading control of Beta-III-tubulin (or equivalent) in Figures 2 and 7 to show that it is a specific reduction in tau and not a general reduction in neuronal protein (which could be reflective of a reduction in neuronal content/complexity).

Reviewer #2 (Remarks to the Author):

This work explores the relationship between pathogenic tau and autophagy in iPSC derived neurons. This question has been extensively studied previously in cellular and animal models, but this work extends to iPSC-derived human neurons. It begins with a characterization of autophagy in a series of patient and unaffected iPSC-derived neural progenitor cells (NPCs) and differentiated neurons (6-8 weeks), showing that autophagy is elevated in the cells from diseased individuals

(Fig 1). This baseline data is used to rationalize a focused chemical screen of molecules that are known to impact autophagy. The screen relies on LysoTracker and CYTO-ID and high content microscopy. After screening, four molecules were subject to dose dependence studies (Fig 2a) and then studied in the wider panel of iPSC-derived cells. The top compounds had stronger activity against insoluble tau (60-90%) than soluble tau (50%), which seems consistent with an autophagic mechanism. The authors then explore the effects of pharmacologic autophagy induction on the stress responsiveness of the iPSC-derived cells. Using Abeta and NMDA as stressors, they find that stress responsiveness was improved (as judged by rescue of cell viability). To confirm these findings, they conduct a series of studies in human ex vivo neurons to show that tau is recruited to the phagolysosomes/lysosomes by the mTor inhibitors. Target validation was also conducted using known mTorc1/2 substrates (p-Akt, p-beclin, etc.; Fig 6). This last point is important, as the inhibitors could be acting through other pathways – but the author's rigorously address this issue. Finally, and perhaps most remarkably, the authors find that these effects are persistent for 12-16 days after washout. Initial PK studies (Fig 7) suggest that compound is retained in cells for a significant period, at near the therapeutic levels. Overall, this is an impressive study, which both confirms previous models of autophagic protection from tauopathy and extends those studies with some interesting PK-PD relationships. The manuscript is also well written and logically presented. Only very minor changes are suggested.

1. In Figure 1F, it would be useful to add more quantitative criteria for triage. What percentage toxicity was allowed? What percentage of compounds passed each filter? A few more details would help the reader understand whether the ultimately identified compounds are unusual.
2. Based on the dose- (Fig 2A-D) and time-dependence (Fig 7B-D), there seems to be a strong relationship between total tau and p-tau. In other words, both total and p-tau are reduced with similar drug effects. It seems worth pointing this out in the discussion, as the autophagy system in this model does not seem to prefer p-tau – just insoluble tau.
3. Why didn't all of the autophagy related molecules in the screen show similar effects? What is special about AZD8055, AZD2014 and OSI-027?

To the Reviewers,

We are grateful for the comments contributing to improvement of our manuscript, which we have revised based upon Reviewers' feedback and requested changes. We have included additional experimental data, altered the text accordingly (changes highlighted using 'Track Changes' in the word document), and have included relevant and requested raw data within the Source Data File. In response to each of the Reviewers' points (in Arial font), we indicate the changes made below (in italic *Times New Roman* font).

As a general note, all western blots in main Figures and Supplementary Figures that show a dotted line were cropped because certain samples, ending up not pertaining to this manuscript, were excluded from the figure. Full, uncropped and unedited western blot images are included within the Source Data File provided, showing that samples and vehicle-controls were always run on the same gel.

Reviewer #1

Some minor points to be addressed:

1. The authors suggest an increase in components of the ALP machinery specifically in mutant vs. control neurons (Figure S1). However, it is hard to appreciate this as each line has a separate, spliced panel – and so difficult to assess whether they are different exposures etc. It would be useful to include samples from each line run on a single gel – e.g. n=3 from each line at 12 weeks on the same gel to allow the reader to appreciate the change and any variability.

The reviewer certainly refers to Supplementary Fig. 2a (not Supplementary Fig. 1), and we are thankful for the feedback. Previously, blots were cropped and rearranged in order to group sample sets by genotype and facilitate comparison between controls and tau variant/mutation carriers. But we agree that this underrepresents experimental accuracy. In reality, both FTD-patient derived neurons (A152T and P301L) and at least one control set were run within the same gel, as allowed per gel size vs. sample number. For each experiment, gels were loaded with equal amount of protein per well (10 µg) and densitometry was normalized to the loading control of each sample in the same blot. This should, in part, correct for variability across blots. Samples distributed across different gels were developed simultaneously, i.e. with the same ECL reagent, equal exposure time and same film. Now, to better represent this setup and follow Reviewer's advice, we have changed Supplementary Fig. 2a to show uncropped blots as much as possible. In addition, uncropped and unedited blot images are included within the Source Data file provided.

Regarding the last part of the Reviewer's comment, ("e.g. n=3 from each line at 12 weeks on the same gel"), we are of the opinion that there is greater value in showing the time-course change in autophagy markers and tau, tested multiple times as biological replicates, rather than showing multiples of a single time-point (e.g. 12 weeks) for all cell lines within one gel. The relative change over time for each marker is informative to reveal genotype-dependent trends. Our conclusions are drawn from comparing the "slope" between curves, which are consistently higher for FTD-derived neurons relative to controls. We avoided extrapolation of relative fold-changes, which would require a different type of analysis.

2. The authors routinely include a side panel of recombinant tau ladder beside the tau western blots, but this is redundant as **1**. It is a spliced lane and not part of the main gel **2**. The samples are not dephosphorylated and so the molecular weight will be a reflection of many factors including phosphorylation state **3**. As a recombinant tau ladder, one assumes the ladder itself is dephosphorylated and therefore wouldn't be detected by the pTau 396 antibody? I would suggest removing the ladder throughout as it doesn't add to the figures or help interpretation.

The recombinant tau ladder was included as a guide to identify monomeric vs. oligomeric tau. But we agree with all the points brought up by the Reviewer and, as such, the recombinant tau ladder panels have been removed from all figures. Instead, tau bands used for densitometry analysis are indicated within brackets.

In answer to point (1), the tau ladder was included in the gel probed for total tau (TAU5 antibody) but, relative to neuronal tau, it required a lower sensitivity ECL reagent and lower exposure time to allow for image resolution of all 6 isoforms bands. Therefore, it was presented as a separate cropped lane.

In answer to point (2), we had only included the recombinant tau ladder as a proxy migration guide to identify monomeric tau, given that neurons expressing endogenous tau isoforms \pm post-translational modifications will reveal multiple bands on western blot.

In answer to point (3), P-Tau antibodies do not stain the recombinant ladder, and the cropped image previously shown had been developed with a total tau antibody.

3. The banding patterns observed with the pTau 396 antibody are highly variable between lines, time points and experiments. For the quantification of these blots, the authors use some higher molecular weight bands - can they provide evidence these bands are tau species and not non-specific – especially as they do not appear to be detected by the Tau5 antibody?

Using Supplementary Fig. 2a as an example (and published data in Silva et al. Stem Cell Reports 2016), the antibody P-Tau S396 detects monomeric phosphorylated tau as well as oligomeric tau of reduced SDS solubility and variable molecular weight. The higher MW bands are usually detected with higher intensity and specificity in patient-derived neurons expressing tau-A152T or tau-P301L (>250 kDa), relative to non-mutant control neurons. Variability in P-tau band pattern is expected across lines with different tau mutations and relative to tau-WT carriers, and we believe these represent early disease-relevant forms of tau. This variability was also expected based on differences between human disease/pathology associated with tau-A152T (a rare risk variant for FTD, AD and synucleinopathies) and tau-P301L (an autosomal dominant mutation associated with familial forms of FTD). Finally, experimental conditions will also contribute to some variability in band patterning, namely the cell lysis and protein purification method (as specified in the Methods section pages 29-30), mainly for those bands that are highly dependent on SDS solubility and originate a smear of oligomeric bands.

As the reviewer pointed out, high MW oligomeric forms of tau are usually not detected by the TAU5 antibody, unless under more native conditions (less denaturing) such as in the cellular fractionation experiment (Fig. 5a, Supplementary Fig. 6a, results section page 16), which suggests

a matter of epitope availability and SDS solubility-dependent denaturation of the protein for antibody binding.

Nonetheless, to address the concern about antibody and western blot bands specificity for tau, we now included new experimental data employing our previously published CRISPR/Cas9-engineered MAPT knockdown (MAPT-kd) lines, generated from the parental line tau-A152T, FTD19-L5-RC6 (Silva et al. Stem Cell Reports 2016). The results of this experiment are shown in the revised version of Supplementary Fig. 2i-k and described in the Results section pages 5-6. Briefly, we used MAPT-Kd neurons and undifferentiated NPCs (no tau expression) from the same line and compared antibody band patterns with those of differentiated neurons of all genotypes (Supplementary Fig. 2k). As stated in the revised version of the manuscript, “Tau-specific bands were present in control neurons, showed increased intensity in FTD neurons and clear reduced intensity in CRISPR MAPT-Kd neurons, and were absent in NPCs. These bands corresponding to monomeric (~50-60 kDa) and oligomeric of \geq 250 kDa tau (brackets, Supplementary Fig. 2j) were employed from here on for western blot densitometry analysis.”

4. The prolonged decrease of tau levels following mTORi is a very interesting finding by the authors. However, the western blots presented in Figure S8 show some variability between compounds and experiments – e.g. pTau396 in the 10M AZD2014 appear to decrease at the later timepoints only, total tau in the OSI-027 treated cells fluctuates a little. I think it would help the reader to have an example western blot for each compound included the main figure, and all 3 replicates as supplementary data.

We thank the reviewer for the advice and opportunity to revise this figure to make it easier for the reader to interpret the results. As suggested, we have moved the representative western blots from Supplementary Fig. 8 into the revised Fig. 7, and additional biological replicate blots are included within the Source Data file. Fig. 7 and Fig. 8 have been re-arranged to accommodate the changes made.

On a technical note, and as specified in the Methods section, the samples used for post-treatment time-course analysis were directly lysed in SDS loading buffer (due to the very large number of samples and replicates), and protein concentration was not quantified. This generates protein loading variability across samples, as shown by differences in actin bands' intensity (see Source Data file for larger images of the original blots). This has to do with inherent variability in cell density per well for each sample, unavoidable for neuronal cultures differentiated for 6 - 8 weeks, rather than a loss in viability, which was also measured. Loading variability is corrected by normalizing tau densitometry to actin densitometry. The remaining “fluctuation” during the time-course of 20 days represents the recovery of tau levels with time, which can be specific for each compound properties.

5. Although the treatments do not appear to reduce neuronal viability, the cells used will still be in a neurodevelopment phase and I wonder if tau reduction could be having more subtle effects on neurodevelopment – for example reduced neurite outgrowth. It would be beyond the scope of the current manuscript to look at this, however it would be good to include this in the discussion and I

think it would also be good to include western blot data with tau, actin, and the additional loading control of Beta-III-tubulin (or equivalent) in Figures 2 and 7 to show that it is a specific reduction in tau and not a general reduction in neuronal protein (which could be reflective of a reduction in neuronal content/complexity).

We hypothesized that the studied mTORi had a lasting effect on tau as a consequence of compound stability, effect on the autophagy pathway and needed time for tau protein to be re-synthesized and “re-accumulate”, rather than a halt in neuronal differentiation and integrity. But the Reviewer’s point is an important one and we are thankful for the suggestion of an additional control.

All samples shown in the revised version of Fig. 7 and Supplementary Fig. 8 (which include 24h treatment at 10 μ M compound equivalent to Fig. 2), i.e. all time-courses post-treatment for A152T, P301L and WT samples have been immunoprobed for β -III-tubulin. The results are described on pages 19-20 and discussed on page 24 of the revised manuscript. Briefly, within the first 24 - 48 hrs after treatment there was a general trend for ~10-20% reduction in β -III-tubulin levels for the highest concentrations of mTORi, but by day 4 and onward levels were back to 100%. Given that reduction of tau persisted for a longer period of at least 12 days, this suggests uncoupling between tau reduction and effect on β -III-tubulin, and that neuronal integrity was not significantly affected as per β -III-tubulin and actin levels over time. The exception was for rapamycin, that caused a 20-30% reduction in β -III-tubulin that persisted through the 20 days of analysis (Supplementary Fig. 8a, d), suggesting that it could have detrimental effects on neurodevelopment over time.

Reviewer #2

Only very minor changes are suggested.

1. In Figure 1F, it would be useful to add more quantitative criteria for triage. What percentage toxicity was allowed? What percentage of compounds passed each filter? A few more details would help the reader understand whether the ultimately identified compounds are unusual.

We thank the Reviewer for the suggestion and opportunity to make the results of the small molecule screen clearer to the reader. Accordingly, we revised Fig. 1f to now include more quantitative information on each step of the screen leading to identification and selection of the mTORi compounds studied. This information is now also included within the Results section on page 7-8.

The compounds studied here were selected, not for being unusual, but based on the following criteria: of the 21 final hits identified in FTD neurons, 12 compounds are predicted to be mTOR pathway modulators, and three of these compounds (AZD8055, AZD2014 and OSI-027) are described as direct mTOR inhibitors (catalytic mTORC1/2 ATP-competitive inhibitors). We selected these compounds that allowed us to build and test a hypothesis for mechanism of action in human ex vivo neurons. Despite the great interest in the field in mTOR function and work in rodent systems, to the best of our knowledge, the studies described here have not been previously performed with mTOR inhibitors in tauopathy patient neurons.

2. Based on the dose- (Fig 2A-D) and time-dependence (Fig 7B-D), there seems to be a strong relationship between total tau and p-tau. In other words, both total and p-tau are reduced with similar

drug effects. It seems worth pointing this out in the discussion, as the autophagy system in this model does not seem to prefer p-tau – just insoluble tau.

This is an important point and result that we initially failed to emphasize in the discussion. In the revised version of the manuscript, this is now included in the Discussion on page 24, stating: “In this ex vivo human neuronal model, mTORi compounds, and therefore autophagy, led to an overall tau reduction, without clear discrimination between total tau and P-tau (Fig. 2a-h, Supplementary Fig. 3a-d), but with preference for tau species of reduced solubility (Fig. 3b-d) predicted to represent misfolded and aggregation-prone tau.”

We interpret this to mean that these compounds are activating the macro-autophagy machinery that preferentially will degrade misfolded and oligomeric/aggregation-prone proteins, rather than a more selective process, e.g. by chaperone-mediated autophagy. This does not exclude the possibility that other tau antibodies, e.g. conformation-specific tau antibodies would not show a different effect from total tau. Determination of the ‘tau species’ targeted for degradation by autophagy is beyond the scope of this paper but, building on the observations presented here, it is something we will continue to explore.

3. Why didn't all of the autophagy related molecules in the screen show similar effects? What is special about AZD8055, AZD2014 and OSI-027?

Please see our answer above to Reviewer’s comment #1 where this point is also addressed. Of the final group of hit compounds from the screen, the ones chosen for this manuscript (the three mTORi) were not selected based on differential or optimal effects on tau, but based on known targets for these small molecules, which allowed us to propose a mechanism of action, and thoroughly dissect our hypothesis.

Nonetheless, the final 21 hits identified to activate autophagy and reduce tau in FTD neurons were not exclusively mTOR inhibitors, nor was it our intention to state that other autophagy activators would not exert a similar effect. mTOR pathway modulators (upstream of mTOR and direct mTOR inhibitors) were the largest class of compounds identified, among other molecules of unknown targets or that have no predicted direct effect on the autophagy pathway. These are the subjects of other ongoing projects aiming to provide a similar level of target validation and mechanistic insight as we have attempted to provide for the mTOR inhibitors in the present manuscript.

REVIEWERS' COMMENTS:

Reviewer #1 (Remarks to the Author):

The authors have addressed all of my comments in full, and I thank them for taking the time to provide a comprehensive rebuttal to each of my points. No further comments other than to complement them on an excellent paper!

Reviewer #2 (Remarks to the Author):

The authors have nicely addressed my concerns, especially around clarification of the selection criteria for compound advancement. The key choices are much more transparent.